# EAPO: Enhancing Policy Optimization with On-Demand Expert Assistance

## Abstract

Large language models (LLMs) have recently advanced in reasoning when optimized with reinforcement learning (RL) under verifiable rewards. Existing methods primarily rely on outcome-based supervision to strengthen internal LLM reasoning, often leading to inefficient exploration and sparse rewards. To mitigate this issue, we propose *Expert-Assisted Policy Optimization* (EAPO), a novel RL framework that enhances exploration by incorporating multi-turn interactions with external experts during training. Unlike prior methods, where policies reason in isolation, EAPO incentivizes the policy to adaptively determine when and how to consult experts, yielding richer reward signals and more reliable reasoning trajectories. External assistance ultimately internalizes expert knowledge into the policy model, amplifying the model's inherent reasoning capabilities. During evaluation, the policy model has been well-optimized to solve questions independently, producing improved reasoning paths and more accurate solutions. Experiments on mathematical reasoning benchmarks, including AIME 2024, AIME 2025, and AIMO 2025, show that EAPO consistently outperforms expert-assisted workflow, expert-distilled models, and RL baselines, with an average gain of 5 points over self-exploratory models.

## 1 Introduction

Large reasoning models (LRMs) have recently achieved remarkable progress in areas such as mathematics and logical reasoning. However, guiding these models to produce accurate multi-step reasoning chains remains a major challenge, as it involves broad search spaces, sparse feedback and unstable optimization process (Ouyang et al., 2022; Christiano et al., 2017; Gou et al., 2023). Existing post-training frameworks typically optimize models with supervised fine-tuning (SFT) and subsequently refine models through reinforcement learning from human feedback (RLHF) to better align model outputs with human preferences (Ouyang et al., 2022). Although effective in various applications, RLHF methods suffer from reward hacking, over-optimization on proxy objectives, and instability during training (Gao et al., 2023; Casper et al., 2023).

Reasoning models are particularly sensitive to these challenges, leading to high exploration costs and unstable behavior when optimized for complex reasoning tasks. Various reward designs are proposed to alleviate these problems: verifiable rewards provide objective reward signals that help alleviate reward hacking (Lambert et al., 2024). Rita et al. (2024) mitigates reward over-optimization by using rewards calibrated with demonstrations, while Barj & Sautory (2024) leverages feedback from LLMs to remedy objective mis-generalization. Although these reward signals are more robust in recent research, existing reinforcement learning (RL) algorithms rely on policy model to explore and exploit on its own to produce long chains of thought, resulting in inefficient exploration and suboptimal optimization. To enhance reasoning performance during evaluation, recent methods incorporates parallel sampling or aggregation of multiple candidate reasoning trajectories (Kumar et al., 2024), such as self-Consistency (Wang et al., 2022), Tree-of-Thoughts (Yao et al., 2023), Mixture-of-Agents (Wang et al., 2024), and LeaP (Luo et al., 2025). Despite improving solution quality, heavy communication cost and implementation complexity are still unsolved and capacity of the policy model is not improved in these test-time scaling based methods.

To address these limitations, we propose *Expert-Assisted Policy Optimization* (EAPO), a novel RL optimization framework that adaptively integrates consulting experts into the model's decision mak-

Figure 1: Framework of EAPO. During training, the policy model adaptively consults experts as assistants. While at test time, model performs reasoning independently without external assistance.

ing process, as illustrated in Figure 1. Specifically, the agent's action space is augmented with a special *consult experts* action, which enables agent improve its reasoning strategy when external assistance is needed. The policy model can therefore choose either to generate the reasoning tokens independently or or to seek a hint from experts. When the *consult experts* action is utilized, the external expert pool provides the assistant responses, prompting the reasoning process of policy model. Additionally, to prevent the agent from over-relying on expert assistance, *consult experts* operation is regarded as a learnable action and optimized jointly with reasoning actions.

During RL optimization, the model is progressively encouraged to rely on its own reasoning ability, ensuring that consultant occurs on demand only when the agent encounters difficult questions or ambiguous reasoning paths. With a curriculum-like optimization design, the agent is allowed to consult experts freely without in the early stages of training, providing richer assistance compared to the sparse feedback from final verifiable rewards. Experimental observations show that policy model gradually learns to consult experts only for challenging questions, rather than relying on assistance for all cases. Moreover, responses from experts help policy model to take deeper reasoning by consolidating multi-perspective insights. This design significantly improves agent's advanced reasoning strategies, such as summarization, judgment and reasoning, which are rarely observed in self-exploratory models. To further strengthen the model's own capability, a penalty term on consultation is introduced. As training progresses, the model is incentivized to consult less frequently, internalizing the consulted reasoning strategies into its own reasoning process. With sufficient training, EAPO effectively internalizes expert knowledge into policy model, eliminating the need for consultation during evaluation.

In our experiments, we comprehensively evaluate three datasets on mathematical reasoning, including AIME 2024, AIME 2025 and AIMO 2025. Compared with various strong baselines, such as expert-assisted workflow, distillation from experts and self-exploratory RL models, EAPO gains consistent improvements on all these benchmarks with a significant improvement of 5 points over self-exploratory RL on averaged evaluation results. Furthermore, training with assistance from experts, policy model achieves lower variance on test sets, indicating greater stability and consistency in its reasoning performance. Our contributions are summarized as follows:

- **Expert-Assisted Optimization:** Expert consultation is designed as a learnable action within the policy, enabling the model to consult assistance on demand during training. This mechanism guides the reasoning trajectory throughout exploration and improves optimization efficiency compared to supervision based solely on final verifiable rewards.

- **Assistance Internalization:** EAPO progressively reduces the reliance on expert assistance throughout training, encouraging the model to internalize expert knowledge into its own reasoning process. During evaluation, the policy model responds entirely independently, yet still demonstrates strong reasoning performance.

- **Enhanced Reasoning Performance:** Extensive experiments on challenging reasoning benchmarks (AIME 2024/2025, AIMO 2025) demonstrate that EAPO achieves superior accuracy and stability. Ablation studies and analyses further show that EAPO consistently improves reasoning performance as expert parallelism and model size scale up.

## 2 EXPERT-ASSISTED POLICY OPTIMIZATION (EAPO)

### 2.1 PROBLEM FORMALIZATION

We consider a complex reasoning task with input $x$ and ground truth answer $g$. A reasoning process can be formalized as a trajectory

$$H_T = \{(\tau_1, \alpha_1, o_1), \ldots, (\tau_t, \alpha_t, o_t), \ldots, (\tau_T, \alpha_T, o_T)\},$$

where $\tau_t$ represents the intermediate reasoning chain, $\alpha_t$ is the policy action that determines whether and how to consult the expert or produce the final answer, and $o_t$ denotes the assistance from experts at step $t$. $T$ represents the final step of the reasoning trajectory. If the policy calls the expert, then $o_t$ corresponds to the expert's feedback; otherwise, $o_t = \varnothing$. Given the current trajectory $H_{t-1} = \{(\tau_i, \alpha_i, o_i,)\}_{i=1}^{t-1}$, the policy model $\pi_\theta$, parameterized by $\theta$, generates $\tau_t$ and $\alpha_t$ at step $t$ as

$$\pi_\theta(\tau_t, \alpha_t \mid H_{t-1}) = \pi_\theta^\tau(\tau_t \mid H_{t-1}) \cdot \pi_\theta^\alpha(\alpha_t \mid H_{t-1}, \tau_t),$$

where $\pi_\theta^\tau$ and $\pi_\theta^\alpha$ denote the probability of intermediate reasoning chain and the consultation decision, respectively. During testing, $\alpha$ is fixed to prevent expert consultation, ensuring that the policy model performs reasoning independently without expert assistance. Therefore, given an input $x$, the probability of generating the entire trajectory can be computed as

$$\pi_\theta(H_T \mid x) = \prod_{t=0}^{T-1} \pi_\theta^\tau(\tau_{t+1} \mid H_t) \cdot \pi_\theta^\alpha(\alpha_{t+1} \mid H_t, \tau_{t+1}) \cdot p(o_{t+1} \mid \alpha_{t+1}),$$

where $H_0 = x$ denotes the initial input, and $p(o_{t+1} \mid \alpha_{t+1})$ is the probability of assistance returned by experts.

Once the full trajectory $H_T$ is generated by $\pi_\theta$, the output answer $\hat{y}$ is extracted from the last reasoning step using a pre-defined extraction rule, *i.e.*, $\hat{y} = \mathcal{E}(H_T)$. The reward function follows an end-to-end verifiable reward design. Given $\hat{y}$ and the ground truth be $g$, the reward is obtained as

$$R = \begin{cases} \mathsf{f1}(\hat{y}, g) & \text{if f1 score is not 0,} \\ 0.1 & \text{if f1 score is 0 with correct format,} \\ 0 & \text{otherwise} \end{cases}$$

Given the training dataset $\mathcal{D}$ consisting of $(x, g)$ pairs, the objective of EAPO can be defined as

$$\max_\theta \mathcal{J}_{\mathsf{EAPO}} = \mathbb{E}_{(x,g)\sim\mathcal{D}, H_T\sim\pi_\theta(\cdot|x)} \left[ R(\mathcal{E}(H_T), g) \right].$$

The key contribution of EAPO lies in introducing an expert assistant signal into the training process, which effectively accelerates and stabilizes optimization. At test time, the model is capable of reasoning independently while retaining the expert knowledge acquired during training, thereby bridging the benefits of expert-guided training and autonomous inference.

### 2.2 MECHANISTIC INTUITION BEHIND EAPO

**Alleviating Sparse Rewards** EAPO enables the policy model to consult expert for assistance at critical steps during training, supplying cues of problem-solving from the external experts. In long-horizon reasoning tasks, merely relying on unguided exploration typically yields few positively rewarded samples. By incorporating external expert assistance, EAPO increases the likelihood of producing partially correct intermediate states and ultimately correct answers, densifying the reward signal, raising the proportion of effective trajectories, and consequently reducing inefficient search. This effect hinges on two assumptions: (i) the expert knowledge is valid for the current task and

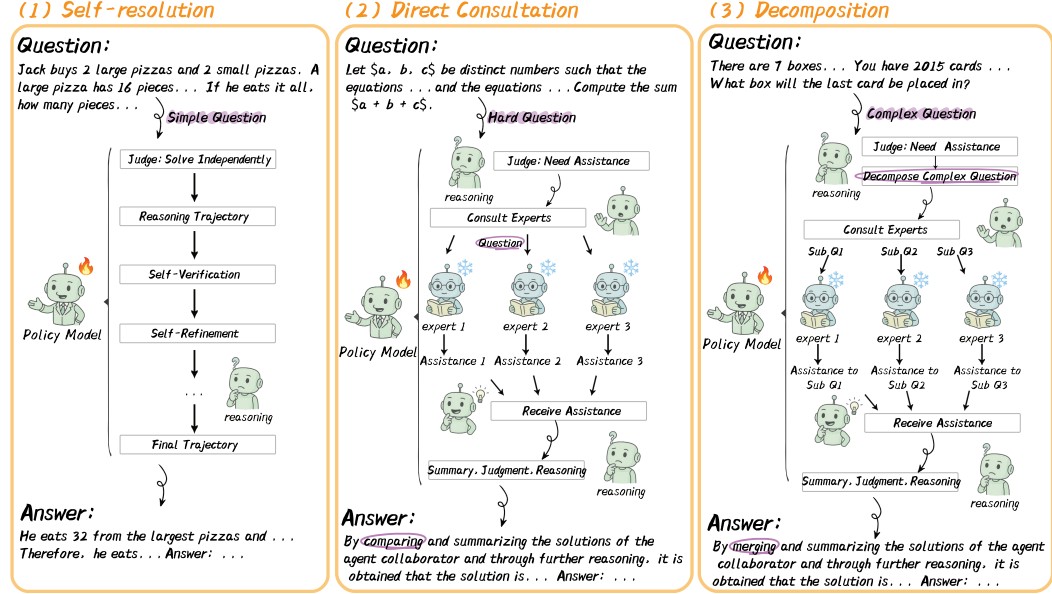

Figure 2: Three rollout paradigms on the training process: (1) Self-resolution (simple): Internal multi-step reasoning without expert assistance; (2) Direct Consultation (hard): Consult three experts directly about the problem and make a comprehensive decision through comparison. (3) Decomposition (complex): Break down into multiple sub-problems, consult experts, and combine the assistance to obtain the final answer.

aligned with the objective, and (ii) the policy model can extract useful features from the assistance and transform it into better decisions, such as selecting more accurate descent directions or shrinking the search space. Under these conditions, EAPO acts as an implicit curriculum, guiding the model more rapidly into regions with high reward and establishing a solid foundation for subsequent incorporation of further external assistance or self-improvement.

**Information Gain**  EAPO can be viewed as a lightweight mechanism of information injection: expert assistance acts as external signals that augment the historical context, enabling policy model to select actions under richer conditioning and thereby tending to reduce decision uncertainty. Although this assistance does not constitute explicit process supervision, it provide local constraints and priors that are relevant to the goal. The expert assistance intuitively aligns the direction of update more closely with the reward structure and lowers the variance of gradient estimates. In other words, once external expert information is injected, the model more readily focuses on factors strongly associated with final target, forms transferable problem-solving patterns, and attains more stable, higher performance with fewer interactions. These advantages likewise hinge on the two conditions noted above. Subsequent experiments and case studies will further explain these mechanistic claims.

## 2.3 Multi-Expert Parallel Reasoning and End-to-End Training

We model the action of asking experts for help as an explicit action available to the agent during training for solving complex tasks. At each step, the policy model $M_{\text{policy}}$ performs private reasoning, may query an expert $M_{\text{expert}}$ for assistance, integrates the returned information into its history, and then either continues the interaction or emits a final answer. Training unfolds in episodes with at most $T$ turns. At each turn, the policy model autonomously decides whether to consult the experts. Three rollout paradigms are observed during training, aligning with different task complexity and patterns of expert use, as shown in Figure 2: self-resolution for simple cases, direct consultation for hard cases, and decomposition for complex cases.

**Rollout Paradigms**  For simple instances, the model solves the task through internal multi-step reasoning without consulting experts. The trajectory comprises private deliberation, iterative re-

finement, and termination with a final answer. Panel (1) in Figure 2 illustrates a representative self-resolution trajectory. It is worth noting that as training progresses, the model likewise adopts this paradigm whenever it has sufficient competence to handle the problem independently.

For harder instances, the model queries three experts on the original problem statement, obtains three independent answers to the same question, and aggregates them via comparison and reconciliation. This strategy improves robustness to single-path errors and reduces reliance on brittle inference chains. Panel (2) in Figure 2 depicts direct consultation and comparison step.

For complex instances, the model first decomposes the task into multiple subproblems, consults experts on each component, and then aggregates the returned evidence into a coherent solution. This paradigm increases coverage, enables error localization, and promotes modular reasoning patterns that transfer across tasks. Panel (3) in Figure 2 presents decomposition, targeted consultations, and evidence fusion. Examples for the three paradigms mentioned above are shown in the appendix E.

**Parallel Evidence Gathering and Test Environment Alignment**    To broaden information coverage within a single turn while keeping computation manageable, we instantiate up to $K$ replicas of the expert model $\{ M_{\mathrm{expert}}^{(i)} \}_{i=1}^{K}$. At turn $t$, $M_{\mathrm{policy}}$ first reasons privately over the history $H_{t-1}$, then may select a concurrency level $C_t$ with $0 \leq C_t \leq K$, constructs structured queries, and dispatches them in parallel:

$$\mathcal{Q}_t = \{ q_{t,i} \}_{i=1}^{C_t}, \quad r_{t,i} = M_{\mathrm{expert}}^{(i)}(q_{t,i}), \quad o_t = \{ r_{t,i} \}_{i=1}^{C_t},$$

where $q_{t,i}$ denotes the query sent to the $i$-th expert at iteration $t$, $r_{t,i}$ denotes the corresponding response returned by the $i$-th expert, and $o_t$ denotes the set of all expert responses collected at iteration $t$. The assistance $o_t$ are integrated to form $H_t$ and drive the next decision or produce the final answer. This design broadens the information gathered in a single turn, improves coverage of reasoning threads, and reduces reliance on any single path.

To promote agent autonomy and reduce reliance on experts, we anneal both the effective acceptance rate of expert calls and the per-episode turn budget during training. Let $s \in \mathbb{N}^+$ denote the global training step and $\rho_s \in (0, 1]$ the probability that an expert response, once returned, is written into history. Early in training, a high acceptance rate accelerates the acquisition of useful patterns from expert feedback. As training progresses, we set $\rho_s = s^{-1}$, gradually suppressing expert access, encouraging internalization of the acquired knowledge, and shifting decision-making to the model's own policy. Concretely, for each expert response we draw $u \sim \mathrm{U}(0, 1)$, where U denotes a sample from the standard uniform distribution. If $u \leq \rho_s$, the response is appended to the history, otherwise it is treated as unavailable and the policy model continues reasoning without external input. In parallel, the episode turn budget is reduced from its initial value toward the evaluation budget, which further promotes concise planning and independent decision making.

## 3 EXPERIMENTS

### 3.1 SETUP

**Models and Baselines**    In this study, we adopt DeepSeek-R1-Distill-Qwen-7B (Guo et al., 2025) (hereafter,7B) as the $M_{\mathrm{policy}}$ and reasoning backbone, and use QwQ-32B (Team, 2025) (hereafter,32B) as the $M_{\mathrm{expert}}$ in heterogeneous collaboration settings. We compare our method against the following baselines: (1) Base Model corresponds to zero-shot inference with 7B, which represents the initial capability. (2) Self Collaboration denotes an offline collaboration workflow in which the policy model is 7B and the experts are three replicas of 7B. (3) Expert-Assisted Workflow refers to a multi-agent workflow where 7B acts as the policy model while three replicas of 32B serve as experts. (4) LeaP (Luo et al., 2025) is a collaborative inference framework in which parallel paths conduct cross-path summarization and routing. (5) Full Parameter (Lv et al., 2023) and LoRA (Hu et al., 2022) distillation represent the full parameter and parameter efficient regimes. (6) Self-Exploratory RL (Yu et al., 2025) retains only outcome-driven reinforcement learning without external experts, requiring the agent to rely entirely on internal reasoning.

Table 1: Evaluation on verifiable tasks. For the RL method, we train the model on DAPO-MATH, while other methods are implemented via offline workflows. We evaluate on AIME 2024, AIME 2025, and AIMO 2025. Specifically, **Pass@32** measures solution accuracy (the higher the better), while **Var** quantifies solution stability (the lower the better). We find that the EAPO consistently outperforms all comparison baselines in terms of mean and variance across all three benchmarks.

| Method | AIME 2024 | | AIME 2025 | | AIMO 2025 | | Avg. | |
|---|---|---|---|---|---|---|---|---|
| | Pass@32↑ | Var↓ | Pass@32↑ | Var↓ | Pass@32↑ | Var↓ | Pass@32↑ | Var↓ |
| Base Model | 51.73 | 0.1112 | 37.19 | 0.0580 | 38.68 | 0.1150 | 42.53 | 0.0947 |
| Self Collaboration | 50.27 | 0.2171 | 35.53 | 0.2004 | 39.68 | 0.2041 | 41.83 | 0.2072 |
| Expert-Assisted Workflow | 58.07 | 0.2137 | 43.16 | 0.1978 | 46.94 | 0.2003 | 49.39 | 0.2039 |
| LeaP | 61.67 | 0.2547 | 39.27 | 0.2165 | 40.31 | **0.0244** | 47.08 | 0.1652 |
| Distillation (full) | 43.54 | 0.0965 | 33.95 | 0.0927 | 28.44 | 0.1169 | 35.31 | 0.1020 |
| Distillation (LoRA) | 51.90 | 0.1056 | 40.02 | 0.0618 | 37.81 | 0.1234 | 43.24 | 0.0969 |
| Self-Exploratory RL | 67.16 | **0.0646** | 55.09 | 0.0625 | 55.22 | 0.0911 | 59.16 | 0.0727 |
| **EAPO (Ours)** | **70.03** | 0.0673 | **58.01** | **0.0564** | **64.17** | 0.0693 | **64.07** | **0.0643** |

Table 2: Effect of expert size and parallelism on reasoning performance. Parallel EAPO consistently outperforms self-exploratory RL and sequential EAPO. In addition, models trained with larger-size experts (32B) surpass those trained with smaller-size ones (14B).

| Method | Parallel | Expert Size | AIME 2024 | | AIME 2025 | | AIMO 2025 | | Avg. | |
|---|---|---|---|---|---|---|---|---|---|---|
| | | | Pass@32↑ | var↓ | Pass@32↑ | var↓ | Pass@32↑ | var↓ | Pass@32↑ | var↓ |
| Self-Exploratory RL | – | – | 67.16 | 0.0646 | 55.09 | 0.0625 | 55.22 | 0.0911 | 59.16 | 0.0727 |
| **EAPO (Ours)** | × | 32B | 68.31 | 0.0673 | 57.02 | 0.0675 | 60.03 | 0.0816 | 61.79 | 0.0721 |
| | √ | 14B | 68.93 | **0.0618** | 54.67 | 0.0693 | 61.06 | 0.0765 | 61.55 | 0.0692 |
| | √ | 32B | **70.03** | 0.0673 | **58.01** | **0.0564** | **64.17** | **0.0693** | **64.07** | **0.0643** |

**Evaluation** We evaluate our method on three mathematical reasoning benchmarks: AIME 2024, AIME 2025, and AIMO 2025. Performance is measured using the mean and variance of accuracy, defined as follows:

- **Mean (Pass@32)**: For each problem, we independently sample 32 final answers and take the arithmetic mean of their F1 as the score for that problem; the dataset score is the mean over problems. Concretely, for problem $q$ with ground-truth answer $y^*$ and the $i$-th prediction $\hat{y}_i$, Pass@32 $= \frac{1}{32} \sum_{i=1}^{32} \text{F1}(\hat{y}_i, y^*)$ and the overall dataset score is $\text{F1} = \frac{1}{|Q|} \sum_{q \in Q} \text{Pass@32}$.

- **Variance (Var)**: Owing to the instability of large model sampling, variance computation serves as an effective measure to assess stability. For each problem $q \in Q$ with ground-truth answer $y^*$, we draw $M = 32$ independent samples and compute $\mu_q = \frac{1}{M} \sum_{i=1}^{M} \text{F1}(\hat{y}_i, y^*)$, where $\mu_q$ is the per-problem mean score (Pass@32). Aggregating over all $|Q|$ problems, we define $\overline{\mu} = \frac{1}{|Q|} \sum_{q \in Q} \mu_q$, $\text{Var} = \frac{1}{|Q|} \sum_{q \in Q} (\mu_q - \overline{\mu})^2$. Thus, $\overline{\mu}$ corresponds to the overall mean score, while $Var$ quantifies the stability across tasks.

## 3.2 EXPERIMENTAL RESULTS

Table 1 reports results on AIME2024, AIME2025, and AIMO2025 (Pass@32 and Var). From Table 1, our method attains the best Pass@32 on all three benchmarks, improving average F1 by 21.54% over the Base Model, 22.24% over Self Collaboration, 14.68% over Expert-Assisted Workflow, 16.99% over LeaP, 20.83% over the two distillation baselines, and 4.91% over self-exploratory RL. These findings further support the effectiveness of EAPO for policy optimization: by invoking expert assistance on demand, it enables more efficient exploration and strengthens LLM reasoning.

**Model Size and Policy Performance** To study the effect of the size of the policy model (Fig. 3), we replace the base model with DeepSeek-R1-distill-Qwen-14B and train from scratch under the same setup, keeping the expert fixed to QwQ-32B. Relative to the 7B model, the 14B model attains higher overall accuracy on the three datasets, but with diminishing marginal returns. Mechanistically, a larger size strengthens internal representations, enabling the model to resolve more difficult

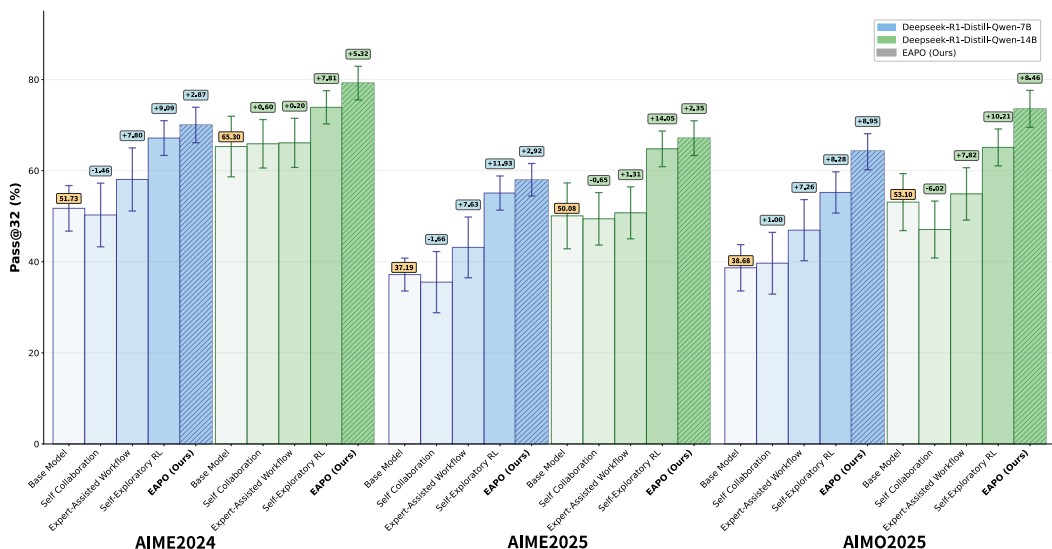

Figure 3: Policy model scaling under fixed expert (QwQ-32B). The model size of policy model is increased from 7B to 14B.

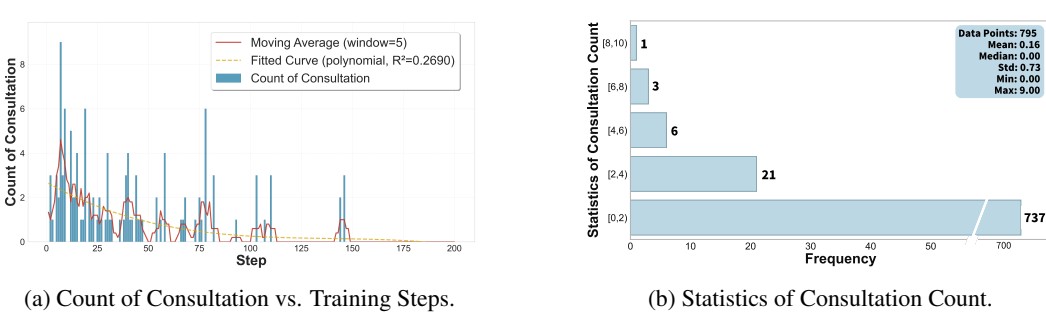

(a) Count of Consultation vs. Training Steps.    (b) Statistics of Consultation Count.

Figure 4: Rate of expert consultation during training. Early training relies heavily on experts; As capabilities are internalized and gates are calibrated, consultation rate of expert decreases, indicating the improvements of reasoning capacity.

cases without expert consultation. However, as hard cases occupy a long-tail minority, incremental gains taper with scale.

**Interplay of Parallelism and Expert Capacity in Performance**  We study two factors influencing the effectiveness of training, as shown in Table 2: the parallelism of expert queries and the model size of the expert model. For parallelism, we compare a sequential regime (one query per round) with a parallel regime (up to three concurrent queries). Parallel querying yields higher solve rates. The most possible explanation is increased information coverage per episode: with a limited number of turns, issuing multiple queries simultaneously surfaces more relevant evidence before the budget expires, rescuing instances that sequential querying fails to resolve due to interaction limits. Performance later in training remains stable as query frequency is annealed, consistent with competence being internalized.

For the model size of the expert, replacing a larger-size expert QwQ-32B (Team, 2025) with a smaller-size one DeepSeek-R1-Distill-14B (Guo et al., 2025) under the same end-to-end RL pipeline yields a measurable drop in performance. The smaller expert provides narrower coverage and noisier guidance, weakening the learning signal. In summary, parallelization primarily improves exploration performance and robustness, whereas the size of the expert model governs external information quality; both dimensions are necessary for stable, high-performing reasoning training.

Table 3: Offline collaboration strategies compared with single-agent baselines. Heterogeneous expert-assisted workflow yields the largest gains for small models and offers minimal marginal benefit for large ones; self-collaboration is inconsistent and can degrade due to redundancy.

| Method | Policy Model | Experts | AIME2024 | AIME2025 | AIMO2025 | Average |
|---|---|---|---|---|---|---|
| Single Model | R1-7B | – | 51.73 | 37.19 | 38.68 | 42.53 |
| | R1-14B | – | 65.30 | 50.08 | 53.10 | 56.16 |
| | QwQ-32B | – | 78.25 | 60.81 | 71.52 | 70.19 |
| Self Collaboration | R1-7B | 7B+7B+7B | 50.27 | 35.53 | 39.68 | 41.83 |
| | R1-14B | 14B+14B+14B | **65.90** | **49.43** | 47.08 | 54.14 |
| | QwQ-32B | 32B+32B+32B | 78.07 | 64.07 | 73.95 | 72.03 |
| Expert-Assisted Workflow | R1-7B | 7B+14B+32B | **55.44** | **42.81** | **46.18** | **48.14** |
| | R1-14B | 7B+14B+32B | 64.65 | 48.65 | **53.98** | 55.76 |
| | QwQ-32B | 7B+14B+32B | **78.37** | **64.63** | **73.95** | **72.32** |

Table 4: Ablation on homogeneous vs. heterogeneous expert pools in EAPO (pass@32 and variance on AIME/AIMO benchmarks). When the expert pool is homogeneous (Deepseek-R1-distill-7B experts for a Deepseek-R1-distill-7B policy), EAPO provides little to no gain over Self-Exploratory RL. In contrast, a heterogeneous expert pool (Llama-8B experts for a 7B policy) consistently improves performance, indicating that complementary capability distributions are more important than sheer expert size.

| Method | Expert | AIME2024 | | AIME2025 | | AIMO2025 | | Average | |
|---|---|---|---|---|---|---|---|---|---|
| | | pass@32↑ | var↓ | pass@32↑ | var↓ | pass@32↑ | var↓ | pass@32↑ | var↓ |
| Self-Exploratory RL | – | 67.16 | 0.0646 | 55.09 | 0.0625 | 55.22 | **0.0911** | 59.16 | **0.0727** |
| homogeneous EAPO | R1-7B | 63.74 | 0.0613 | 54.73 | **0.0570** | 58.08 | 0.1084 | 58.85 | 0.0756 |
| heterogeneous EAPO | Llama-8B | **67.19** | **0.0604** | **55.92** | 0.0651 | **58.90** | 0.0927 | **60.66** | **0.0727** |

**Dynamics of Call Rate and Capability Internalization**   To quantify reliance on the expert and its evolution in the training phase, we track the call rate of the expert, as shown in Figure 4. The policy model calls experts frequently in early training to obtain solution cues, but the number of calls declines markedly as training proceeds. Even with a large reduction in call rate, final performance remains stable, evidencing the joint effects of capability internalization and gating calibration.

**Conditional Expert-Assisted Workflows**   To isolate the effect of expert heterogeneity in collaboration, we design an offline workflow (Table 3) and compare: (1) Single Agent, (2) Self Collaboration, and (3) Heterogeneous Expert-Assisted Workflow. The results reveal clear potentials and limits. Self collaboration yields no consistent gains and can degrade performance. We hypothesize a concrete mechanisms, which is redundancy amplification, highly correlated responses inflate context length without adding information, diluting attention and crowding out salient evidence.

By contrast, heterogeneous collaboration is most beneficial when policy model is small. Mutual-information gain is probably a driver that diverse experts contribute complementary priors and coverage. As policy model grows stronger, these benefits taper and can reverse, as weaker experts may introduce noisy or contradictory rationales. In summary, heterogeneous experts help chiefly by increasing usable information, whereas model size of experts governs response quality.

**Evolution of Response Length and Reasoning Style**   Figure 5 presents the evolution of response length when training with EAPO compared to self-exploratory RL. In the early phase, EAPO responses are noticeably shorter. This reduction can be attributed to expert querying: by

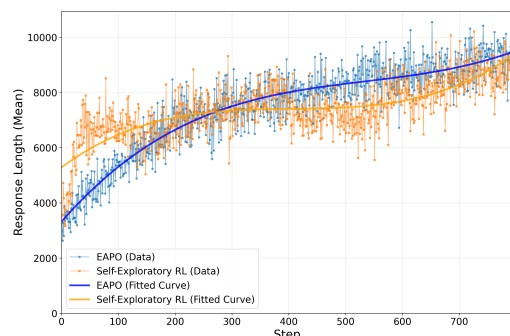

Figure 5: Dynamics of response length over training (compare EAPO with Self-Exploratory RL)

Table 5: Offline workflow evaluation on non-mathematical benchmarks. Results show that expert assistance consistently improves over the single-agent baseline, indicating that expert-augmented workflows remain beneficial beyond mathematical reasoning.

| Method | Policy Model | Experts | HumanEval | HLE | GPQA | MMLU | EvalPlus | HotpotQA | simpleQA |
|---|---|---|---|---|---|---|---|---|---|
| Single Model | R1-7B | – | 86.79 | 7.00 | 45.92 | 48.16 | 83.72 | 12.86 | 6.70 |
| | R1-14B | – | 93.90 | 7.47 | 56.14 | 56.82 | 94.57 | 30.08 | 8.95 |
| | QwQ-32B | – | 59.96 | 8.34 | 59.67 | 67.03 | 60.61 | 34.09 | 12.31 |
| Expert-Assisted Workflow | R1-7B | 7B+7B+7B | **84.76** | 6.69 | 44.23 | 45.01 | 80.49 | 11.93 | 5.74 |
| | R1-7B | 14B+14B+14B | 83.54 | 7.27 | 46.06 | 52.59 | **80.91** | 27.85 | 8.27 |
| | R1-7B | 32B+32B+32B | 66.46 | **7.29** | **48.18** | **54.96** | 65.15 | **29.45** | **11.13** |

Table 6: Generalization of EAPO from math-only training to non-mathematical benchmarks. EAPO consistently outperforms both the base model and a self-exploratory RL baseline across code and scientific tasks, indicating that expert-assisted policy optimization learned on math transfers effectively to other structured reasoning domains.

| Method | HumanEval | HLE | GPQA | MMLU | EvalPlus | HotpotQA | simpleQA | Average |
|---|---|---|---|---|---|---|---|---|
| Base Model | 86.79 | 7.00 | 45.92 | 48.16 | 83.72 | 12.86 | 6.70 | 41.59 |
| Self-Exploratory RL | 89.34 | 15.96 | 48.13 | 55.47 | 85.16 | 21.46 | 14.56 | 47.15 |
| **EAPO (Ours)** | **91.56** | **16.07** | **49.92** | **58.61** | **88.04** | **23.75** | **16.82** | **49.25** |

delegating segments of the reasoning process to external guidance, the model suppresses unnecessary exploratory text. As training progresses, the EAPO curve gradually exceeds self-exploratory RL. A plausible interpretation is that, after internalizing expert guidance, the model learns to construct more structured and explanatory chains of reasoning while improving performance. Importantly, this expansion in response length is not mere verbosity.

**Homogeneous and heterogeneous expert pools.** We further ablate the structure of the expert pool (Table 4). When the experts are *homogeneous* to the policy (Deepseek-R1-distill-7B experts for a Deepseek-R1-distill-7B policy), EAPO's overall performance fluctuates around the pure-RL baseline. Together with our self-collaboration results shown in Table 1, this suggests that a homogeneous expert pool provides little information gain and may even amplify redundancy. In contrast, with a *heterogeneous* expert pool (Llama-8B experts for a 7B policy), EAPO consistently outperforms pure RL on all three benchmarks. We attribute this to differences in capability distributions between Deepseek-R1-distill-7B and Llama-8B: EAPO can exploit this heterogeneity to realize complementary strengths between policy and experts, leading to more effective policy optimization.

**Robustness and cross-domain generalization.** We further evaluate EAPO and expert-assisted workflows on a suite of non-mathematical benchmarks, including HumanEval (Chen, 2021), HLE (Phan et al., 2025), GPQA (Rein et al., 2024), MMLU (Hendrycks et al., 2020), EvalPlus (Liu et al., 2023; 2024), HotpotQA (Yang et al., 2018), and simpleQA (Wei et al., 2024). First, we construct offline collaboration workflows and compare self-collaboration with the Expert-Assisted Workflow. As shown in Table 5, adding expert models consistently improves performance on these non-math tasks, indicating that expert-assisted collaboration remains useful beyond mathematical reasoning. Second, to assess the generalization ability of EAPO, we fix the training set to the math-only DAPO-MATH corpus and evaluate on the same non-math benchmarks. Table 6 shows that, even when trained solely on mathematical reasoning data, EAPO still yields stable gains over both the base model and the self-exploratory RL baseline, highlighting the robustness and cross-domain generalization of our approach.

## 4 RELATED WORK

**Expert-Guided Distillation** Expert-guided distillation aims to accelerate learning by coupling reinforcement objectives with expert demonstrations. The core idea is that a student policy can consult an expert in uncertain states, imitate its behavior, and gradually reduce reliance as competence improves. Existing studies typically ground this idea in two forms. The first is *teacher-guided RL*, where a fixed teacher supervises the student and the student may eventually surpass the teacher in

performance (Schmitt et al., 2018; Biré et al., 2024; Zhao et al., 2024). The second is *offline distillation*, where the learner regresses onto a static teacher using pre-collected trajectories (Rusu et al., 2015). More recent work emphasizes interactive imitation and selective querying, allowing expert signals to be injected at pivotal states to reduce distribution shift (Ross et al., 2011; Xie et al., 2022; Chi et al., 2020; Buck et al., 2017). Despite these advances, two challenges remain: (i) most methods assume persistent or externally scheduled reliance on experts, which risks over-dependence, and (ii) offline fitting cannot correct expert biases or adapt to environment-specific rewards. Our approach addresses both. We learn an explicit invocation policy that decides when to query the expert during on-policy interaction, so that supervision is obtained precisely where it is most useful. Crucially, the invocation rate is optimized end-to-end to converge to zero, ensuring that dependence vanishes as the student matures. This formulation both mitigates distribution shift and transforms expert guidance into a self-contained capability, yielding a policy model that is ultimately independent.

**Hierarchical Reinforcement Learning** Hierarchical RL (HRL) tackles long-horizon problems by decomposing control (Sutton et al., 1999; Bacon et al., 2017; Vezhnevets et al., 2017; Rashid et al., 2020; Yu et al., 2022). Progress in this area has produced two dominant paradigms. One jointly trains options, termination conditions, and high-level controllers (Bacon et al., 2017), while another adopts a manager–executor structure where high-level goals condition low-level execution (Vezhnevets et al., 2017). These designs improve exploration but maintain permanent dependence on higher-level directives during both training and deployment. By contrast, our method treats the expert as a temporary external capability. Early in training, the expert is frequently invoked to stabilize learning, but as competence grows, invocations diminish until disappear. This yields two advantages: (i) we inherit the sample efficiency benefits of hierarchical guidance in the early phase, and (ii) we guarantee that long-term execution becomes fully independent, avoiding the burden of permanent hierarchical control.

**Self Distillation and Social Learning** Self-distillation and related paradigms exploit a policy's own historical behaviors to bootstrap improvement. One line of work shows that a student distilled from its earlier versions can surpass the original policy under certain conditions (Furlanello et al., 2018). Self-imitation further reinforces validated behaviors by replaying high-return trajectories (Oh et al., 2018). Social learning demonstrates that complex strategies can be transferred between agents and eventually consolidated into individuals capable of independent execution (Ndousse et al., 2021; Vinyals et al., 2019). However, self-distillation often assumes unlimited replay of past trajectories and social learning presumes multiple agents remain available for transfer. Our work differs by a principled mechanism that ensures guidance accelerates early learning but disappears once competence is achieved, guaranteeing autonomy without needing auxiliary teachers indefinitely.

## 5 CONCLUSION

In this paper, we propose Expert-Assisted Policy Optimization (EAPO), a novel reinforcement learning framework designed for incorporating external assistance signal. The consultation of experts, which is a key component in EAPO, is leveraged as a learnable action solely during training, allowing policy model adaptively determines when and how to consult experts to improve the exploration strategy. Across three challenging mathematical reasoning benchmarks, EAPO outperforms both expert-assisted evaluation workflow and self-exploratory RL models. Extensive experiments reveal that: 1) Incorporating expert assistance during RL training significantly improves policy optimization with better reasoning paths and more accurate solution strategies. 2) Scaling expert assistance, through parallel signals or larger expert models, consistently enhances performance, demonstrating that richer external signals during training can effectively strengthen the optimization process. 3) Analysis across different policy model sizes shows that consulting experts can also provide improvements as the student model capacity increases.

Although EAPO achieves promising results, the present study has certain limitations that open opportunities for further exploration: i) We will explore more external signals from not only experts, but also various environments to further improve the reasoning abilities of policy models. ii) Systematically studies on cross-task generalization will be conducted to evaluate the generalization of EAPO. iii) Incorporating evolution based method as external signals into RL to to further advance multi-round reasoning agents.

## REPRODUCIBILITY STATEMENT

We have undertaken extensive efforts to ensure the reproducibility of our work.

- Models and Baselines: Section 3.1 provides comprehensive descriptions of all models and baselines employed in our experiments, including implementation details and configuration settings.

- Evaluation Protocols: Section 3.1 also details the benchmarks and metrics used for evaluation, enabling performance comparisons under consistent conditions.

- Hyperparameter Settings and Resources: Appendix A reports the key hyperparameters used during training and provides direct download links to all models and datasets, ensuring transparency and accessibility.

- Training Procedures: Appendix B describes the complete training workflow, including potential looping logic during optimization, to clarify the exact dynamics of the process.

- System Prompts: Appendix C contains the full set of system prompts used in both training and testing phases, preserving experimental conditions that may affect model behavior.

- Expert Model Deployment: Appendix D documents the deployment of the expert model as an on-demand inference service via vLLM. We describe how the service is exposed through a standardized HTTP/OpenAI interface. In addition, we provide the exact serving script to ensure faithful replication of our deployment environment.

- Code Availability: We will publicly release the entire codebase, including training scripts, evaluation pipelines, and configuration files, to further support faithful reproduction and extension of our work.

We believe that the combination of detailed documentation, explicit hyperparameter reporting, resource sharing, and forthcoming code release provides sufficient transparency for researchers to replicate and validate our findings.

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

## A  IMPLEMENTATION DETAILS

The pre-trained policy model of DeepSeek-R1-Distill-Qwen-7B can be download via `https://huggingface.co/deepseek-ai/DeepSeek-R1-Distill-Qwen-7B` and DeepSeek-R1-Distill-Qwen-14B can be download via `https://huggingface.co/deepseek-ai/DeepSeek-R1-Distill-Qwen-14B`. The pre-trained expert model of QwQ-32B can be download via `https://huggingface.co/Qwen/QwQ-32B`. The training dataset DAPO-Math-17K is available at `https://huggingface.co/datasets/BytedTsinghua-SIA/DAPO-Math-17k`, and the evaluation datasets of AIME2024 and AIME2025 can be download on `https://huggingface.co/math-ai`, and AIMO2025 can be download on `https://www.kaggle.com/competitions/ai-mathematical-olympiad-progress-prize-2`. Table 7 lists the key hyperparameters used in self-exploratory RL and EAPO.

Table 7: Critical hyperparameters distilled from the training script.

| Hyperparameter | Value | Description |
|---|---|---|
| Max prompt length | 2048 | Input context window for prompts |
| Max response length | 20480 | Maximum generated tokens per sample |
| Train batch size | 32 | Number of prompts per training step |
| Gen batch size | 96 | Number of prompts per sampling batch |
| Learning rate | 1e-6 | Optimization step size |
| LR warmup steps | 10 | Steps to ramp LR up to target |
| Clip Ratio (Low) | 0.20 | Policy update stability window |
| Clip Ratio (High) | 0.28 | Policy update stability window |
| Temperature | 1.0 | Sampling stochasticity |
| Top-p | 1.0 | Nucleus sampling cutoff |
| Top-k | $-1$ | vLLM backend (no explicit top-k limit) |
| Sequence parallel (Ulysses) | 4 | Long-sequence efficiency via partitioning |
| Tensor model parallel | 4 | Inference/model-parallel scaling |

## B  INTERACTION PROCESS DURING TRAINING

This section provides a concrete illustration of how the agent interacts with experts during training. The example shows a typical episode: the model first performs private deliberation and decomposes the task into sub problems, then dispatches structured queries to multiple experts in parallel, collects their replies, and integrates the returned evidence into the evolving history. When uncertainty remains, the model issues follow up checks before synthesizing the final answer.

The interaction trace also clarifies two important design choices. First, expert queries and returns are serialized in a simple, reproducible schema that makes the control flow explicit and auditable. Second, the policy model is encouraged to use experts not only for initial exploration but also for evidence confirmation. Together, these choices enable broad information acquisition within a turn while maintaining a disciplined loop of reasoning, consultation, and consolidation.

```
Model-Expert Loop

Model think:
<think>I decompose the problem into two subproblems.</think>

Model calls:
<agent_calls>[
  {"expert_id":1,"input_parameters":{"query":"Sub-question A"}},
  {"expert_id":2,"input_parameters":{"query":"Sub-question B"}}
]</agent_calls>

Model receives:
<agent_returns>[
  {"expert_id":1,"result":"Answer A."},
  {"expert_id":2,"result":"Answer B."}
]</agent_returns>
```

```
Model think:
<think>Let me confirm whether my previous reasoning and the answer given by the
    expert are correct.</think>

Model calls:
<agent_calls>[
  {"expert_id":1,"input_parameters":{"query":"Check-query A"}},
  {"expert_id":2,"input_parameters":{"query":"Check-query B"}},
  {"expert_id":3,"input_parameters":{"query":"Check-query C"}}
]</agent_calls>

Model receives:
<agent_returns>[
  {"expert_id":1,"result":"Response A."},
  {"expert_id":2,"result":"Response B."},
  {"expert_id":3,"result":"Response C."}
]</agent_returns>

Model think:
<think>With answers A and B and responses A, B, and C confirming the answers and
    previous thinking, I got the final answer.</think>

Final:
Answer: ...
```

## C  PROMPTS DURING TRAINING AND EVALUATION

The training prompt grants the policy model the ability to coordinate experts, specify queries in a structured format, and integrate returned assistance into its own reasoning.

The evaluation prompt is deliberately minimal and focuses the policy model on producing a single, final answer in a standardized format. This design suppresses auxiliary process signals, aligns behavior with the evaluation environment, and enables fair comparison across methods and datasets. In combination, the training and evaluation prompts support broad information acquisition during learning while encouraging autonomy resoning at test time.

**Prompts during Training**

```
You are a helpful assistant that can solve a given question step by step. To do so,
    you can consult with a panel of three experts. Each question you can ask
    experts 10 times totally.

You can ask one or more experts for their opinion. You can ask them the same
    question or different questions simultaneously. The experts will provide their
    answers, which you can use to form your final response. The agent call query
    and results are enclosed within <agent_calls> </agent_calls> and <agent_returns
    > </agent_returns> tags respectively.

Agent calls query should be formatted as a list of dictionaries.
For each agent call (an item in the list), you need to provide these keys:
    (1) expert_id (int): Required. The ID of the expert you want to consult. Can be
    1, 2, or 3.
    (2) input_parameters (dict): Required. All input parameters for the expert, in
    dictionary format.

Here are the experts you can use:
Description:
This is a panel of three auxiliary intelligent agents (ID 1, 2, 3) that you can
    consult when you are unsure about the answer to a question. The agents'
```

```
        responses may not always be correct, but they can offer you alternative
        perspectives to help you think further.

    Parameters:
        (1) query (str): Required. The query string describing the question you want to
        ask or discuss.
        (2) The answer from the agent, including thinking process.

    The reasoning process is wrapped by <think></think>.
    The final answer begins with "Answer: ".

    For example:
    User: What is the capital city of NomanLand and what is its main export?
    Assistant:
    <think>
        This question has two parts. I will ask two different experts to answer each
         part. I'll ask expert 1 about the capital and expert 2 about the main export.
    </think>
    <agent_calls>
        [{"expert_id": 1, "input_parameters": {"query": "capital city of NomanLand"}},
        {"expert_id": 2, "input_parameters": {"query": "main export of NomanLand"}}]
    </agent_calls>
    <agent_returns>
        [{"expert_id": 1, "result": "The capital city of NomanLand is NomanCity."},
        {"expert_id": 2, "result": "The main export of NomanLand is widgets."}]
    </agent_returns>
    <think>
        I see. Expert 1 says the capital is NomanCity, and Expert 2 says the main export
         is widgets.
    </think>
    Answer: The capital of NomanLand is NomanCity and its main export is widgets.
```

**Prompts during Evaluation**

```
You are a helpful assistant.

You must give the final answer in the last line, in the format of "Answer: xxx", xxx
    represents the final answer.
```

## D  vLLM SERVER

We deploy the expert model as an on-demand inference service through vLLM. The service is exposed to the outside world through a standardized HTTP/OpenAI interface, supports long-context reasoning, and can be efficiently scaled in a multi-GPU environment. The training end maps the query set constructed in each round into a request and sends it to the server. After the request is returned, it is written into the history, thus supporting the parallel training process from information acquisition to output decision.

**Inference Service through vLLM**

```
vllm serve ${MODEL_PATH} \
    --port ${PORT} \
    --host :: \
    --gpu-memory-utilization 0.8 \
    --max-model-len 32768 \
    --tensor-parallel-size 8 \
    --served-model-name expert \
    --trust-remote-code \
```

```
        --disable-log-requests
```

## E CASE STUDY

Below are three sets of cases corresponding to the three rollout paradigms in Section 2.3. All are derived from real interaction records during the training phase (minimized for readability). They illustrate three collaborative modes: self-resolution, direct consultation, and decomposition. In the first case, the model relies entirely on internal multiple step reasoning to arrive at an answer; in the second, the model consults multiple experts concurrently on the same problem, forming a comprehensive judgment through cross comparison; and in the third, the model first breaks a complex task into several sub-problems, poses targeted questions to the experts, and gradually integrates the evidence to ultimately synthesize a global solution.

---

**Self Resolution**

```
User: Let $a, b, c$ be distinct numbers such that the equations $x^2 + ax + 1 = 0$
    and $x^2 + bx + c = 0$ have a common real root, and the equations $x^2 + x + a
    = 0$ and $x^2 + cx + b = 0$ also have a common real root. Compute the sum $a +
    b + c$.

Assistant:
<think>
    Step 1. Let the common root of the first pair be r:
        r^2 + ar + 1 = 0,   r^2 + br + c = 0
        (a-b)r + (1-c) = 0
        r = (c-1)/(a-b).
    Step 2. Let the common root of the second pair be s:
        s^2 + s + a = 0,   s^2 + cs + b = 0
        (1-c)s + (a-b) = 0
        s = (b-a)/(1-c).
    Step 3. Combine the two:
        rs = ((c-1)/(a-b)) * ((b-a)/(1-c)) = 1
        s = 1/r.
    Step 4. Substitute s = 1/r into s^2 + s + a = 0:
        (1/r^2) + (1/r) + a = 0
        1 + r + a r^2 = 0.
        From r^2 + ar + 1 = 0
        r^2 = -ar - 1. Substituting:
        1 + r + a(-ar - 1) = 0
        r(1 - a^2) + (1 - a) = 0.
        If a != 1, then r = (a-1)/(1-a^2) = -1/(1+a).
    Step 5. Substitute r = -1/(1+a) back into r^2+ar+1=0:
        (1/(1+a)^2) - a/(1+a) + 1 = 0
        (1+a)^2 + 1 - a(1+a) = 0   a+2=0   a=-2.
    Step 6. Then r = -1/(1-2) = 1. Plugging into x^2+bx+c=0 gives 1+b+c=0
        b+c=-1.
    Step 7. Therefore a+b+c = -2 + (-1) = -3.
</think>
Answer:-3
```

---

**Direct Consultation**

```
User: The integers from $1$ to $n$ are written, one on each of $n$ cards. The first
    player removes one card. Then the second player removes two cards with
    consecutive integers. After that, the first player removes three cards with
    consecutive integers. Finally, the second player removes four cards with
```

consecutive integers. What is the smallest value of $n$ for which the second
player can ensure that he completes both his moves?

Assistant:
```
<think>
    Provide three distinct approaches (interval-splitting, hitting-set of 4-blocks,
    and fragmentation/invariant), then synthesize them to obtain the minimal n.
</think>
<agent_calls>
    [
      {"expert_id": 1, "input_parameters": {"query": "Find the minimal n
    guaranteeing the second player can remove 2 consecutive and later 4 consecutive
    , regardless of the first player's moves."}},
      {"expert_id": 2, "input_parameters": {"query": "Recast the game as a
    hitting-set/covering problem on all 4-consecutive windows; derive threshold n
    ."}},
      {"expert_id": 3, "input_parameters": {"query": "Use fragmentation invariants (
    remaining length = 8 after three moves) to characterize when a 4-run must
    remain; produce a constructive strategy."}}
    ]
</agent_calls>
<agent_returns>
    {"expert_id": 1, "result": "Interval-splitting method. After the first player
    removes a single card at position $p$, the remaining integers form two
    contiguous segments of lengths $A=p-1$ and $B=n-p$ (so $A+B=n-1$). The second
    player's 2-consecutive removal inserts a 2-gap inside one of these segments,
    splitting it into two parts whose lengths sum to (chosen segment) $-2$. If,
    before the first player's 3-consecutive removal, we can ensure at least two
    segments have length $\ge 4$, then no single 3-block can destroy both; hence a
     4-block will survive for the final move. Sufficiency for $n=14$: Since $A+B=13
    $, either $\min(A,B)\le 3$, which forces $\max(A,B)\ge 10$, letting us
    split the larger segment into two parts both $\ge 4$ by removing two cards
    near its middle (since $10-2=8=4+4$); or else $\min(A,B)\ge 4$ and then $\
    max(A,B)\ge 9\ge 6$, so by removing two from the larger side we can keep one
    part $\ge 4$ while the other side already contributes another $\ge 4$. Thus
    after our move there are at least two $\ge 4$ segments; the opponent's 3-block
     can hit at most one of them, leaving a 4-run intact. Necessity: For $n=13$,
    pick $p=4$ so $(A,B)=(3,9)$. Whatever 2-consecutive the second removes, after
    that there is at most one segment of length $\ge 4$. The first player then
    deletes three consecutive from the length-9 segment, centered to split it into
    $(3,3)$, leaving no 4-run. Therefore $n$ must be at least $14$.", "status": "
    SUCCESS_AGENT"}
    {"expert_id": 2, "result": "Hitting-set viewpoint on 4-windows. Consider the
    family of all 4-consecutive windows W_i = {i,i+1,i+2,i+3} for i=1,...,n-3. The
    second player's final success is equivalent to the existence of a window W_i
    fully contained in the remaining set after three moves (sizes removed: 1, then
    2, then 3 -> total 6 removed). To prevent a 4-run, the six removed numbers must
     hit every W_i (i.e., form a 4-cover).\n\nWhen n=13, the first player can force
     such a 4-cover: remove {4} first, and no matter which two consecutive the
    second removes, the first can choose a 3-block (e.g., {6,7,8} in a typical case
    ) so that every 4-window intersects one of the removed positions. Hence the
    second cannot guarantee a 4-run at the end. For n=14, the second can always
    choose his 2-consecutive so that there exist at least two disjoint 4-windows
    not simultaneously hittable by a single 3-block; thus after the opponent's move
    , at least one 4-window survives. This yields the threshold n=14.", "status": "
    SUCCESS_AGENT"}
    {"expert_id": 3, "result": "Fragmentation invariant & constructive plan. After
    the first three moves, exactly 6 numbers are removed and 8 remain. To eliminate
     all 4-runs, the opponent must fragment the 8 survivors into pieces of maximum
    length <= 3 (e.g., 3+3+2 or 3+2+2+1). There are only three gaps available (
    sizes 1,2,3 from the three deletions), so the second player's objective in his
    first move is to place the 2-gap so that, regardless of where the later 3-gap
    is placed, the maximum surviving piece after that removal is >= 4.\n\nA
```

```
        constructive strategy at n=14: Observe the split (A,B) with A+B=13. If min(A,B)
         <= 3, then the other side has length >= 10; remove two near its middle to
        create two subsegments of lengths >= 4 each. If both sides are >= 4, then the
        larger side is >= 9; remove two within the larger side so that one subpiece
        remains >= 4 while the smaller original side (>= 4) is untouched. In both
        subcases, after the second player's move there are two disjoint pieces of
        length >= 4, and a single 3-block cannot destroy them both, ensuring a 4-run
        survives. For n=13, the adversarial split (3,9) defeats this plan, as any 2-gap
         yields at most one piece >= 4, which can be broken by a centered 3-removal
        into pieces of size <= 3.", "status": "SUCCESS_AGENT"}
</agent_returns>
<think>
    All three perspectives agree on the same threshold. The interval-splitting
    method formalizes a guaranteed way for the second player, at n=14, to leave two
     >=4 segments before the final 3-removal; a single 3-block cannot eliminate
    both. The hitting-set view explains why n=13 fails: the first player can force
    a 4-cover with six deletions. The fragmentation invariant reframes the goal as
    preventing all pieces from being of length <=3 and yields a simple constructive
     rule for where to place the 2-gap. Thus the smallest n for which the second
    player can ensure completing both moves is 14.
</think>
Answer: 14
```

## Decomposition

```
User: Find all positive integers $n$ for which all positive divisors of $n$ can be
      placed into the cells of a rectangular table under the following constraints:
- Each cell contains a distinct divisor.
- The sums of all rows are equal.
- The sums of all columns are equal.

Assistant:
<think>
    Decompose the task:
    (A) Structural necessities: if the divisors of n fill an m*k grid ($mk=\tau(n)$)
    , then row/column sums must be $\sigma(n)/m$ and $\sigma(n)/k$ (integers), so
    $m \mid \sigma(n)$, $k \mid \sigma(n)$; also rule out $1\times k$ or $k\times 1
    $.
    (B) Exclude any $2\times k$ case for $n>1$ by showing equal column sum forces
    each column to be $(x,\, n+1-x)$, which is impossible with distinct divisors.
    (C) Handle $\tau(n)$ odd ($n$ a perfect square): argue that $m,k\geq 3$ leads to
     a contradiction via extremal-sum bounds; conclude no solution for $n>1$.
</think>
<agent_calls>
    [
      {"expert_id": 1, "input_parameters": {"query": "Derive necessary divisibility/
      size constraints on m,k from the equal-row/column-sum conditions; handle 1*k
      and k*1 edge cases."}},
       {"expert_id": 2, "input_parameters": {"query": "Prove impossibility for any 2*
      k arrangement with all divisors when n>1 (pair-sum argument leading to S = n+1)
      ."}},
       {"expert_id": 3, "input_parameters": {"query": "Treat $\tau(n)$ odd (n is a
      perfect square): show no m*k with m,k>=3 can work for n>1 using extremal
      averages/paired-divisor bounds; provide a consistency check."}}
    ]
</agent_calls>
<agent_returns>
    {"expert_id": 1, "result": "Let $\\tau(n)$ be the number of positive divisors
    and $\\sigma(n)$ their sum. If the divisors fill an $m\\times k$ rectangle,
    then $mk=\\tau(n)$, each row sums to $\\sigma(n)/m$, and each column sums to $
```

```
    \\sigma(n)/k$; hence $m\\mid\\sigma(n)$ and $k\\mid\\sigma(n)$. If $m=1$ (or $k
    =1$), then all column (row) sums must be equal across singleton columns (rows),
     forcing all entries equal---impossible with distinct divisors unless $\\tau(n)
    =1$, i.e., $n=1$. So for $n>1$ we must have $m,k\\ge 2$.", "status": "
    SUCCESS_AGENT"}
    {"expert_id": 2, "result": "Rule out $2\times k$ for $n>1$. In a $2\times k$
    table with equal column sums, every column must consist of a pair whose sum is
    a constant $S$. Since the set is all divisors of $n$, one column contains $n$,
    so $S-n$ must be a positive divisor; thus $S>n$ and hence $S\geq n+1$. On the
    other hand, for any divisor $d$, we have $d + \tfrac{n}{d} \leq n+1$ with
    equality iff $\{d,\, n/d\}=\{1,\,n\}$. Therefore the only way to have every
    column sum equal to a common $S\geq n+1$ is to set $S=n+1$ and make every
    column the pair $(1,\,n)$. That is impossible once $k\geq 2$ because there is
    only one $1$ and one $n$. Hence no $2\times k$ arrangement exists for $n>1$ (
    and similarly no $k\times 2$).", "status": "SUCCESS_AGENT"}
    {"expert_id": 3, "result": "Treat $\tau(n)$ odd $\Rightarrow n$ is a perfect
    square and any rectangular factorization must have $m,k$ odd. The smallest
    nontrivial case is $3\times 3$. Let the divisors be $1=d_1<d_2<\cdots<d_9=n$.
    Column sums must all equal $S_c=\sigma(n)/3$. The column containing $n$ has sum
     at least $n+d_1+d_2=n+1+d_2$. Any column avoiding $n$ has sum at most $d_7+d_8
    +d'_9$ where $d'_9\leq d_8<n$; by the paired-divisor bound, the average of any
    pair $(d,\,n/d)$ is $\leq (n+1)/2$, implying $\sigma(n)/\tau(n)\leq (n+1)/2$
    and hence $S_c=\sigma(n)/3\leq 3(n+1)/2$. Comparing the column with $n$ versus
    one without forces $S_c\geq n+1+d_2$ and simultaneously $S_c\leq 3(n+1)/2$; for
     $n>1$ these bounds require each non-$n$ column to include an element $\geq (n
    +1)/2$. But there are at most two divisors $\geq (n+1)/2$ other than $n$ itself
     (namely $\lceil \sqrt{n}\rceil$ and its complement), while there are at least
    two non-$n$ columns in a $3\times 3$ grid, and for larger odd $m,k$ the
    shortage worsens. Hence no $m,k\geq 3$ arrangement exists for $n>1$. This is
    consistent with computational checks for $n\leq 60$, which find no example
    beyond $n=1$.", "status": "SUCCESS_AGENT"}
</agent_returns>
<think>
    Synthesis: From (1) we reduce to $m,k\geq 2$ for $n>1$. From (2) any arrangement
     with $\min(m,k)=2$ is impossible. Thus if $n>1$, we would need $m,k\geq 3$;
    but (3) shows this cannot happen when $\tau(n)$ is odd (squares), and even $\
    tau(n)$ already fails by (2). Therefore the only $n$ that works is $n=1$ ($\tau
    =1$; the $1\times 1$ table trivially satisfies all conditions).
</think>
Answer: 1
```

# F  THE USE OF LARGE LANGUAGE MODELS (LLMs)

We used LLMs solely as writing aids to polish language (e.g., grammar, wording, and clarity) for text written by the authors. All suggestions from LLMs were reviewed by the authors and accepted, modified, or discarded at our discretion. No passages were inserted without author verification, and all references and factual claims were curated and checked by the authors.

The authors take full responsibility for the contents of this paper, including any text refined with LLM assistance. LLMs are not eligible for authorship and are not listed as authors. No non-public or sensitive data were provided to the LLM service beyond manuscript text, and our usage adhered to relevant institutional and venue policies on LLM use.

