# OpenReview forum: "EAPO: Enhancing Policy Optimization with On-Demand Expert Assistance"
_ICLR.cc/2026/Conference — Submitted to ICLR 2026_

### Official Review · Reviewer_Ka6W · 2025-10-25

**Soundness:** 2
**Presentation:** 3
**Contribution:** 2
**Rating:** 6
**Confidence:** 3

**Summary:**

The paper introduces Expert-Assisted Policy Optimization (EAPO), a reinforcement learning (RL) framework designed to enhance the reasoning capabilities of large language models (LLMs) in complex tasks, particularly mathematical reasoning. EAPO augments the policy model's action space with a "consult experts" action, allowing the model to seek assistance from external experts (stronger LLMs) during training on-demand. This is optimized end-to-end with verifiable rewards, incorporating mechanisms like parallel expert querying, acceptance rate annealing, and consultation penalties to internalize knowledge and reduce reliance on experts over time. During evaluation, the policy reasons independently. Experiments on AIME 2024/2025 and AIMO 2025 show EAPO outperforming baselines like self-exploratory RL, expert-assisted workflows, and distillation methods, with gains of ~5 points on average accuracy and improved stability.

**Strengths:**

1.	EAPO creatively treats expert consultation as a learnable action within the RL policy, enabling adaptive, on-demand guidance during training. This addresses key RL challenges in reasoning tasks: sparse rewards and inefficient exploration.
2.	The experiments are comprehensive, evaluating on challenging math benchmarks with Pass@32 and variance metrics. EAPO consistently outperforms baselines (e.g., +4.91% over self-exploratory RL on average). Ablations on expert parallelism (up to K=3), expert size (14B vs. 32B), and policy scaling (7B to 14B) provide insightful evidence of benefits.
3.	The paper emphasizes reproducibility with detailed setups (e.g., models like DeepSeek-R1-Distill-Qwen-7B as policy, QwQ-32B as expert), hyperparameters (Table 4), prompts (Appendix C), and vLLM deployment (Appendix D).

**Weaknesses:**

1.	While effective on AIME/AIMO, the evaluation is confined to mathematical tasks with verifiable rewards. Generalization to other reasoning domains (e.g., code, commonsense, or multi-modal) is not explored, despite claims of "complex reasoning." The paper acknowledges this in limitations but lacks even preliminary cross-task experiments. This narrows the claimed impact on "large reasoning models (LRMs)."
2.	EAPO's gains hinge on high-quality experts (e.g., 32B > 14B in Table 2). If experts are noisy or biased, the policy might internalize errors, exacerbating reward hacking— a known RLHF issue cited but not deeply mitigated here. Ablations on noisy experts or homogeneous vs. heterogeneous pools could strengthen robustness claims. Additionally, the fixed expert pool (up to 3 replicas) limits diversity; integrating more varied experts (e.g., specialized for sub-tasks) could be discussed.
3.	Training involves multi-turn interactions with experts, potentially increasing costs (e.g., parallel queries but up to T=10 turns). While annealing reduces this, no direct comparison of training efficiency (e.g., wall-time or FLOPs) vs. baselines is provided. For larger policies/experts, scalability might be challenging, especially since vLLM is used but not benchmarked against alternatives.
4.	The mechanistic intuitions (Section 2.2) on alleviating sparse rewards and information gain are intuitive but lack formal proofs or bounds (e.g., on variance reduction or convergence). Related work connections (e.g., to HRL or self-distillation) are solid, but EAPO's novelty could be sharper by quantifying how it improves over them (e.g., via regret bounds).

**Questions:**

1.	Could you provide results on non-math tasks (e.g., coding or planning) to assess generalization? The limitations mention cross-task studies—any preliminary data?
2.	How sensitive is EAPO to expert quality? E.g., what if experts are weaker than the policy or provide conflicting advice?
3.	Did you experiment with dynamic expert pools (e.g., routing to specialized experts) or more than K=3 parallelism?
4.	The reward function includes a 0.1 partial credit for correct format but wrong answer—how does this affect optimization vs. pure 0/1 rewards?
5.	In Appendix E cases, how were examples selected? Are there failure modes where EAPO over-consults or under-internalizes?

---

> ### Author Response · Authors · 2025-11-20
> **To Reviewer Ka6W (1/6)**
>
> Dear Reviewer Ka6W,
>
> Thank you for your thorough review and constructive feedback. All your concerns have been carefully responded. The manuscript is carefully revised accordingly. We sincerely hope our responses fully address your questions.
>
> **To major concerns:**
>
> **Q1:** *While effective on* *AIME/AIMO, the evaluation is confined to mathematical tasks with verifiable rewards. Generalization to other reasoning domains (e.g., code, commonsense, or multi-modal) is not explored, despite claims of "complex reasoning." The paper acknowledges this in limitations but lacks even preliminary cross-task experiments. This narrows the claimed impact on "large reasoning models (LRMs)." Could you provide results on non-math tasks (e.g., coding or planning) to assess generalization? The limitations mention cross-task studies—any preliminary data?*
>
> **A1:** Thank you for this comment. To better assess the generality of EAPO beyond mathematical reasoning, we have added a series of experiments on non-math benchmarks, including HumanEval, HLE, GPQA, MMLU, EvalPlus, HotpotQA, and simpleQA, which cover code generation, scientific reasoning, and knowledge-intensive multi-step reasoning tasks. Our analysis has two parts:
>
> 1. Expert-assisted workflows on non-math datasets:
>
> We first evaluate self-collaboration and the Expert-Assisted Workflow in an offline setting. We observe that introducing an expert model into the workflow yields consistent and significant performance gains, even for code and science tasks. This indicates that the idea of “expert-assisted reasoning” is not limited to mathematics, but can also extend to other domains that require structured reasoning.
>
> | Method  | Policy Model | Experts | HumanEval  | HLE  | GPQA  | MMLU | EvalPlus  | HotpotQA  | simpleQA | Average |
> | ------ | ---- | --- | ---- | ---- | ---- | ---- | ----- | ---- | --- | ------- |
> |                          |              |         | Code Generation | Scientific Reasoning | Scientific Reasoning | Scientific Reasoning | Code Generation | Knowledge-Intensive Multi-Step Reasoning | Scientific Reasoning | Non-Mathematical Benchmark |
> | Base Model               | R1-7B        | --      | 86.79           | 7.00     | 45.92                | 48.16                | 83.72           | 12.86                                    | 6.70                 | 41.59                      |
> | Base Model               | R1-14B       | --      | 93.90           | 7.47     | 56.14                | 56.82                | 94.57           | 30.08                                    | 8.95                 | 49.70                      |
> | Base Model               | QwQ-32B      | --      | 59.96           | 8.34                 | 59.67                | 67.03                | 60.61           | 34.09                                    | 12.31                | 43.14                      |
> | Expert-Assisted Workflow | R1-7B        | R1-7B   | 84.76           | 6.69                 | 44.23                | 45.01                | 80.49           | 11.93                                    | 5.74                 | 39.84                      |
> | Expert-Assisted Workflow | R1-7B        | R1-14B  | 83.54           | 7.27                 | 46.06                | 52.59                | 80.91           | 27.85                                    | 8.27                 | 43.78                      |
> | Expert-Assisted Workflow | R1-7B        | QwQ-32B | 66.46           | 7.29     | 48.18                | 54.96                | 65.15           | 29.45                                    | 11.13                | 40.37                      |
>
> 2. Generalizing EAPO from math-only training to evaluations on non-math tasks:
>
> To directly evaluate the generalization of EAPO, we use only the mathematical-reasoning dataset (dapo-math) for training. We find that, even solely trained  on mathematical reasoning data, EAPO consistently outperforms Self-Exploratory RL on these non-math benchmarks. This demonstrates that expert-assisted policy optimization is a broadly applicable approach rather than one constrained to mathematical reasoning.
>
> |                     | HumanEval | HLE   | GPQA  | MMLU  | EvalPlus | HotpotQA | simpleQA | Average |
> | ------------------- | --------- | ----- | ----- | ----- | -------- | -------- | -------- | ------- |
> | Base Model          | 86.79     | 7.00  | 45.92 | 48.16 | 83.72    | 12.86    | 6.70     | 41.59   |
> | Self-Exploratory RL | 89.34     | 15.96 | 48.13 | 55.47 | 85.16    | 21.46    | 14.56    | 47.15   |
> | EAPO                | 91.56     | 16.07 | 49.92 | 58.61 | 88.04    | 23.75    | 16.82    | 49.25   |
>
> Taken together, these results show that: (i) expert-assisted collaborative workflows remain effective in non-math domains, and (ii) the policy trained with EAPO on math data exhibits transferable and stable gains on other structured reasoning tasks. This provides empirical support for our claim that EAPO generalizes beyond purely mathematical reasoning.

---

> ### Author Response · Authors · 2025-11-20
> **To Reviewer Ka6W (2/6)**
>
> **Q2:** *EAPO's gains hinge on high-quality experts (e.g., 32B > 14B in Table 2). If experts are noisy or biased, the policy might internalize errors, exacerbating reward hacking— a known* *RLHF* *issue cited but not deeply mitigated here. Ablations on noisy experts or homogeneous vs. heterogeneous pools could strengthen robustness claims. Additionally, the fixed expert pool (up to 3 replicas) limits diversity; integrating more varied experts (e.g., specialized for sub-tasks) could be discussed. How sensitive is EAPO to expert quality? E.g., what if experts are weaker than the policy or provide conflicting advice?*
>
> **A2:** Thank you for this detailed and insightful comment. In the revised version, we expand our analysis of expert quality and expert-pool structure along several dimensions to better address your concerns about robustness and sensitivity.
>
> 1. Homogeneous vs. heterogeneous expert pools: sensitivity to expert strength
>
> We first add ablations comparing homogeneous and heterogeneous expert pools:
>
> **Homogeneous expert pool (expert = policy).** When we set the expert to be exactly the same as the policy model, EAPO performs slightly worse than Self-Exploratory RL on AIME2024/2025 and slightly better on AIMO2025, with overall differences within noise. Together with our self-collaboration findings, this indicates that when experts and policy have essentially the same capability, EAPO provides little information gain, and redundant signals may even introduce mild noise.
>
> **Heterogeneous expert pool (different abilities).** When using Deepseek-R1-distill-Qwen-7B as the policy and Llama-8B as the expert, EAPO consistently outperforms Self-Exploratory RL on AIME2024, AIME2025, and AIMO2025. We attribute this to complementary ability distributions, not just model size, allowing EAPO’s on-demand querying mechanism to effectively leverage heterogeneous expertise and learn stronger exploration and reasoning strategies.
>
> |                     | Expert                 | AIME2024 | AIME2024 | AIME2025 | AIME2025 | AIMO2025 | AIMO2025 | Average | Average |
> | ------------------- | ---------------------- | -------- | -------- | -------- | -------- | -------- | -------- | ------- | ------- |
> |                     |                        | pass@32  | var      | pass@32  | var      | pass@32  | var      | pass@32 | var     |
> | Self-Exploratory RL |                        | 67.16    | 0.0646   | 55.09    | 0.0625   | 55.22    | 0.0911   | 59.16   | 0.0727  |
> | homogeneous EAPO    | Deepseek-R1-distiil-7B | 63.74    | 0.0613   | 54.73    | 0.0570   | 58.08    | 0.1084   | 58.85   | 0.0756  |
> | heterogeneous EAPO  | Llama-8B               | 67.19    | 0.0604   | 55.92    | 0.0651   | 58.90    | 0.0927   | 60.66   | 0.0727  |
>
> 2. Noisy and biased experts: robustness in extreme settings
>
> To further evaluate the robustness of EAPO under noisy or low-quality experts, we design two additional experiments:
>
> **High-error experts.** For each problem in AIME2024/2025 and AIMO2025, we build an “error expert” by selecting 3 incorrect trajectories from 32 QwQ-32B samples (if fewer than 3 are incorrect, we sample from all 32). This yields error rates above 50% (53.33% for AIME2024, 63.33% for AIME2025 and AIMO2025). We then evaluate DeepSeek-R1-distill-Qwen-7B on this collaboration dataset and compare it against an Expert-Assisted Workflow built from 3 randomly sampled trajectories.
>
> **Completely meaningless guidance.** In constructing collaboration trajectories, we replace all expert messages with fixed, content-free text (e.g., “I cannot offer you any assistance; please resolve the issue yourself.”), simulating experts that are essentially uninformative.
>
> These settings simulate scenarios where the expert provides only weak or even purely noisy signals, and allow us to examine the stability and failure modes of EAPO under low-quality experts.
>
> |                      | AIME2024 | AIME2024 | AIME2025 | AIME2025 | AIMO2025 | AIMO2025 | Average | Average |
> | -------------------- | -------- | -------- | -------- | -------- | -------- | -------- | ------- | ------- |
> |                      | pass@32  | var      | pass@32  | var      | pass@32  | var      | pass@32 | var     |
> | Random sampling      | 58.07    | 0.2137   | 43.16    | 0.1978   | 46.94    | 0.2003   | 49.39   | 0.2039  |
> | Error sampling       | 53.73    | 0.2025   | 37.54    | 0.2157   | 37.18    | 0.2076   | 42.82   | 0.2086  |
> | meaningless guidance | 49.50    | 0.1906   | 40.05    | 0.2013   | 39.21    | 0.1940   | 42.92   | 0.1953  |

---

> ### Author Response · Authors · 2025-11-20
> **To Reviewer Ka6W (3/6)**
>
> 3. Fixed expert pool, diversity, and specialized experts
>
> You also point out that a fixed pool of up to three experts may limit diversity, and that integrating more varied or specialized experts could be beneficial. This is an excellent direction.
>
> In the current work, our design goal is to avoid hard-coding, which expert is good at which sub-task or domain. Instead, we let the policy learn during training, which experts to call and how to aggregate their feedback. For this reason, we focused primarily on whether the strategy model can learn how to invoke expert models and whether the strategy can internalize expert capabilities, rather than exploring large, specialized expert pools.
>
> As part of our ongoing and future work, we plan to investigate whether combining multiple heterogeneous experts with complementary strengths can further improve performance within the EAPO framework.
>
> 4. Weaker experts and conflicting advice: current scope and observations
>
> Regarding your question about experts weaker than the policy, we have so far systematically evaluated two cases: (1) experts that are identical to the policy; (2) experts that are stronger and heterogeneous.
>
> The homogeneous experiments already indicate that when experts are not clearly stronger than the policy, EAPO brings little or even slightly negative benefit. We therefore did not further explore scenarios where experts are strictly weaker than the policy, as this is outside the main scope of our current work.
>
> For conflicting expert advice, in our offline workflow experiments, we observe cases where all experts produce different but incorrect final answers; yet the model can still read their reasoning chains, compare them, spot the flaws, and derive the correct solution. This shows that in EAPO or workflow-style setups, the model is not simply “voting” over expert outputs, but mainly exploiting their natural-language reasoning as signals. Even when experts disagree or are all wrong, their reasoning traces remain useful, which underscores a key advantage of EAPO over scalar process supervision: language-based guidance provides a much richer structure than a single process score.
>
> ```Plain
> System Prompt:
>
> Role Declaration
> You function as a collaborative hub agent, forming an equal partnership with Experts A/B/C/D in a cooperative network. You are responsible for integrating your own knowledge with that of the other experts. Your tasks include:  \n1. Scanning knowledge anchors  \n2. Verifying and supplementing knowledge to refine the reasoning process  \n3. Executing the final decision and providing the answer to the question  \n\n
>
> Input Data
> Question: Consider a process which outputs a random English letter with uniform probability (i.e., each with probability 1/26). What is expected time until a sequence \"TENETENET\" appears?  \n
>
> Expert Proposals
> Expert A: Exact Answer: 5,429,503,678,976 letters (approximately 5.43 trillion letters)\n\nConfidence: 95%\n\nI'm highly confident in this approach as it's a well-established principle in probability theory. The only minor uncertainty comes from potential edge cases or overlapping patterns, but in this case, they don't affect the fundamental calculation.  \n
> Expert B: Exact Answer: \\(542951556296\\)\nConfidence: \\(100\\%\\)  \n
> Expert C: Exact Answer: \\(26^9\\)\nConfidence: \\(95\\%\\)  \n
> Expert D: Exact Answer: Approximately \\( 3195168 \\) steps.\n\nConfidence: 90%  \n\n
>
> ======================================================================================
>
> Response:
>
> Explanation: After analyzing all expert inputs, I conclude that this problem requires careful consideration of pattern overlaps in a sequence of independent random letters. Experts A and C suggest a simple approach using 26^9, while Expert B provides the most rigorous analysis by considering self-overlaps of the pattern \"TENETENET\". The pattern has self-overlaps of lengths 5 and 1, which affects the expected waiting time. Expert B's approach using the formula E(X)=26^9 + 26^5 + 26^1 is the most accurate as it accounts for these overlaps. Expert D's answer appears to be incorrect by several orders of magnitude. The correct calculation must account for: 1) The base probability of the full sequence (26^9), 2) The probability contributions from self-overlaps of length 5 (\"TENET\") and length 1 (\"T\"). Using the self-overlap principle and the sum 26^9 + 26^5 + 26^1, we get 5429503678976 + 11881376 + 26.
>
> Exact Answer: 5429515560378 ✅
> Confidence: 98%
>
> ======================================================================================
>
> Ground Truth: 5429515560378
> ```

---

> ### Author Response · Authors · 2025-11-20
> **To Reviewer Ka6W (4/6)**
>
> **Q3:** *Training involves multi-turn interactions with experts, potentially increasing costs (e.g., parallel queries but up to T=10 turns). While annealing reduces this, no direct comparison of training efficiency (e.g., wall-time or FLOPs) vs. baselines* *is* *provided. For larger policies/experts, scalability might be challenging, especially since* *vLLM* *is used but not benchmarked against alternatives.*
>
> **A3:** Thank you for this comment. To analyse training-time efficiency and scalability, we clarify our analysis from several angles:
>
> 1. Training-time cost vs. performance trade-off:
>
> Compared to Self-Exploratory RL, EAPO requires roughly 1.4× training time in our experiments. This extra cost mainly comes from invoking the expert model during training to obtain higher-quality trajectories and feedback. However, at test time, we completely disable expert calls, so the inference-time cost of EAPO is the same as that of Self-Exploratory RL.
>
> 2. Per-step training time:
>
> In our experiments, EAPO’s training time is approximately 5.92 minutes per step with a hardware configuration of 8×8 H20 GPUs.
>
> 3. Training efficiency metrics (EAPO vs. Self-Exploratory RL):
>
> To compare training efficiency more systematically, we introduce three standard learning-dynamics metrics to quantify how fast and how effectively EAPO and Self-Exploratory RL improve under a given training budget:
>
> - TTT@90% (Time-To-Threshold) [1–3]: Let $R(k)$ denote the average reward at step $k$, with total training steps $K$, and final performance $R_{\rm final} = R(K)$. We define the threshold $R_{\rm thresh} = 0.9 \times R_{\rm final}$. Let $k^{\*} = \min\{\,k \in \{1,2,\dots,K\} : R(k) \ge R_{\rm thresh}\,\}$ , and if no such $k$exists, set $k^* = K$. Then $\text{TTT@90\%} = k^*$, which measures how many training steps are needed to reach 90% of the final performance.
> - nAUCN_NN (normalized Area Under the learning Curve) [4,5]: For a given horizon $N$ ($1 \le N \le K$), we approximate $AUC_{N} =\sum_{k=1}^{N-1} \frac{R(k) + R(k+1)}{2}$ , and normalize it as $nAUC_{N} = \frac{ AUC_{N} }{\, N \times R_{\rm final} \,}\;$. When $N = K$, we obtain $nAUC_{K}$, which reflects how “lifted” the reward curve is over the whole training process and thus how efficiently the training steps are used.
> - EIS@10% (Early Improvement Speed) [1,6]: Let $K_{10} = \left\lfloor 0.10 \times K \right\rfloor\;$denote the first 10% of training steps. On $k = 1,2,\dots,K_{10}$, we fit a linear model $R(k) = a + b \cdot k + \varepsilon_{k}\;$, and use the slope $b = \frac{\mathrm{d}R}{\mathrm{d}k}$ as the $\mathrm{EIS@10\%}$ metric, measuring how quickly performance improves in the early stage.
>
> Using these metrics, we show in the revised paper that, despite having somewhat higher per-step cost, EAPO achieves smaller TTT@90%, higher $nAUC_{K}$, and larger EIS@10% than Self-Exploratory RL, indicating faster effective convergence.
>
> |                 | trade                 | Baseline (R1‑7B) | Collaborator (R1‑7B+32B) | Δ%     |
> | --------------- | --------------------- | ---------------- | ------------------------ | ------ |
> | TTT@90% (steps) | the lower the better  | 42               | 33                       | -27.3% |
> | nAUC_N          | the higher the better | 0.5524           | 0.7937                   | +30.4% |
> | EIS@10% (slope) | the higher the better | 0.00553          | 0.01079                  | +95.3% |
>
> [1] Jordan S M, Chandak Y, Cohen D, Zhang M, Thomas P S. Evaluating the Performance of Reinforcement Learning Algorithms[C]//Proceedings of the 37th International Conference on Machine Learning (ICML), Vienna, Austria, 2020.
>
> [2] Azran G, et al. Contextual Pre-planning on Reward Machine Abstractions for Few-Shot Transfer[C]//Proceedings of the International Conference on Automated Planning and Scheduling (ICAPS), 2024.
>
> [3] Taylor M E, et al. Improving Reinforcement Learning with Human Input[C]//IJCAI 2018.
>
> [4] Patterson A, Neumann S, White M, White A. Empirical Design in Reinforcement Learning[J]. Journal of Machine Learning Research, 2024, 25:1-63.
>
> [5] D’Oro P, et al. SAMPLE-EFFICIENT REINFORCEMENT LEARNING[EB/OL]. (OpenReview), 2022.
>
> [6] Thomas M, O’Boyle N M, Bender A, et al. Augmented Hill-Climb increases reinforcement learning efficiency for language-based de novo molecule generation[J]. Journal of Cheminformatics, 2022, 14:68.

---

> ### Author Response · Authors · 2025-11-20
> **To Reviewer Ka6W (5/6)**
>
> 4. Scalability with larger policies/experts and vLLM:
>
> Regarding scalability for larger models, our implementation uses vLLM as the inference backend for experts. Public benchmarks and community usage indicate that vLLM has been successfully deployed for 70B-scale and even ~400B-scale models (e.g., Llama 3.1–405B models), and scales well in both single-model scaling and multi-replica deployment [7–11]. In our framework, the choice of inference engine is modular:
>
> - For open-source experts, we use vLLM to support highly parallel expert querying.
> - For closed-source experts, EAPO can similarly call them through APIs as remote experts.
>
> Thus, from a systems perspective, EAPO can directly inherit the scalability properties of modern high-throughput inference engines.
>
> [7] vLLM Team, “Optimization and Tuning,” vLLM Documentation, 2025. Available: https://docs.vllm.ai/en/latest/configuration/optimization/
>
> [8] vLLM Team, “Announcing Llama 3.1 Support in vLLM,” vLLM Blog, Jul. 23, 2024. Available: https://blog.vllm.ai/2024/07/23/llama31.html
>
> [9] K. Kale, “LLM Compressor is Here: Faster Inference with vLLM,” Red Hat Developers Blog, Aug. 14, 2024. Available: https://developers.redhat.com/articles/2024/08/14/llm-compressor-here-faster-inference-vllm
>
> [10] AWS Neuron Team, “Tutorial: Scaling LLM Inference with Data Parallelism on Trn2,” AWS Neuron Documentation, 2024. Available: https://awsdocs-neuron.readthedocs-hosted.com/en/latest/libraries/nxd-inference/tutorials/trn2-llama3.3-70b-dp-tutorial.html
>
> [11] AMD ROCm Team, “vLLM V1 Performance Optimization,” ROCm Documentation, 2025. Available: https://rocm.docs.amd.com/en/latest/how-to/rocm-for-ai/inference-optimization/vllm-optimization.html
>
> &nbsp;
>
> **Q4:** *The mechanistic intuitions (Section 2.2) on alleviating sparse rewards and information gain are intuitive but lack formal proofs or bounds (e.g., on variance reduction or convergence). Related work connections (e.g., to HRL or self-distillation) are solid, but EAPO's novelty could be sharper by quantifying how it improves over them (e.g., via regret bounds).*
>
> **A4:** Thank you for this comment. The “mechanistic intuitions” in Section 2.2 are mainly intended to explain the design motivation and observed behavior of EAPO, rather than to serve as a full theoretical treatment. Our primary contribution is to propose a new “learning from assistance/observation” policy optimization paradigm and to systematically demonstrate its effectiveness on complex reasoning tasks; a more complete theoretical analysis is a natural direction for future work.
>
> Existing RL policy optimization methods still rely largely on sparse rewards or advantage estimates and make limited use of rich auxiliary or observational signals, which is particularly problematic for long-horizon and high-difficulty reasoning. EAPO directly targets this gap by (i) treating external assistance as an actively requested information-gain channel rather than a fixed hierarchy or static teacher, and (ii) allowing the policy itself to decide when and in what form to invoke assistance and integrate it into learning. This yields a dynamic, decision-driven use of assistance that differentiates EAPO from HRL and self-distillation. Empirically, across multiple tasks and model scales, EAPO consistently improves optimization efficiency and reasoning quality over strong baselines (including HRL and self-distillation), and these gains align well with our intuitions about sparse-reward alleviation and information gain, providing strong practical support for the proposed mechanism.
>
> Finally, we believe that incorporating such auxiliary signals into RL training will become an important component of future RL optimization pipelines and significantly enhance the effectiveness of policy optimization.

---

> ### Author Response · Authors · 2025-11-20
> **To Reviewer Ka6W (6/6)**
>
> **Q5:** *Did you experiment with dynamic expert pools (e.g., routing to specialized experts) or more than K=3 parallelism?*
>
> **A5:** Thank you for this comment. In the current work, we do not explicitly route to “specialized experts” (e.g., manually assigning experts to domains), for two main reasons:
>
> 1. Expert usage is learned, not hard-coded. EAPO is designed so that the policy *learns* via RL when and how to query different experts, rather than following fixed rules like “expert A for math, expert B for code.” Expert calls are on demand, and the choice of which expert to query in a given state is entirely driven by the reward signal, yielding an implicit, learned form of dynamic routing.
> 2. Specialized expert pools for future work. As ongoing and future work, we plan to explore more heterogeneous expert sets (different architectures, pretraining distributions, or task specializations) and study how EAPO can exploit this diversity so that the policy gradually learns to prefer certain experts for certain problem types.
>
> Regarding parallelism, we systematically evaluate K=1 (sequential) vs. K=3 (parallel) querying. K=3 clearly outperforms K=1: with a fixed interaction budget, parallel queries increase information coverage by gathering multiple complementary signals at once, reducing the risk that a purely sequential strategy misses crucial reasoning paths. It is reasonable to expect that larger K (e.g., K>3) could further improve performance at additional compute cost, but since our focus is on the EAPO framework and expert quality/structure, we did not exhaustively sweep higher K in this work.
>
> &nbsp;
>
> **Q6:** *The reward function includes a 0.1 partial credit for correct format but wrong answer—how does this affect optimization vs. pure 0/1 rewards?*
>
> **A6:** Thank you for this question. The effect of the format-related partial reward can be summarized as follows:
>
> First, giving partial credit for correct output format is consistent with prior RL work on complex reasoning and structured outputs, where a small format term helps the model quickly learn how to interact with the environment/tools, rather than only optimizing for the final answer. In our case, assigning a 0.1 reward when the format is correct, but the answer is wrong, serves two roles:
>
> 1. Faster acquisition of interaction patterns: early in training, correct answers are rare, but well-formed outputs (proper reasoning/answer markers, correctly wrapped expert queries, etc.) yield a small, stable positive signal, helping the policy quickly learn how to structure its outputs and call experts properly.
> 2. Mitigating extreme reward sparsity: while the main reward still comes from answer correctness, the small format reward alleviates the “all-zero” regime at the beginning of training, making optimization more stable and reducing the risk of policy collapse.
>
> Second, for evaluation, we mainly use F1 rather than a strict 0/1 accuracy, and the 0.1 format reward is only a minor component on top of the primary task reward. It acts as a stabilizer that encourages reliable output structure, but does not dominate or fundamentally alter the optimization objective.
>
> [1] Li K, Zhang Z, Yin H, et al. WebSailor: Navigating Super-human Reasoning for Web Agent[J]. arXiv preprint arXiv:2507.02592, 2025.
>
> [2] Chen M, Sun L, Li T, et al. Learning to reason with search for llms via reinforcement learning[J]. arXiv preprint arXiv:2503.19470, 2025.
>
> &nbsp;
>
> **Q7:** *In Appendix E cases, how were examples selected? Are there failure modes where EAPO over-consults or under-internalizes?*
>
> **A7:** Thank you for this question. The examples in Appendix E are taken from three typical behavioral patterns we repeatedly observed during training and log inspection; for each pattern, we show representative cases rather than cherry-picking only the best outcomes.
>
> Regarding the potential failure modes:
>
> 1. Over-consulting experts. Even early in training, the policy does not blindly query experts on simple problems and often solves them independently. As training progresses, the annealing schedule further reduces the expert-call probability, so the model increasingly relies on its own reasoning. Both the overall consultation rate and qualitative examples suggest no systematic “over-consulting” behavior.
> 2. Under-internalization. EAPO’s goal is to *internalize* expert knowledge, not rely on experts at test time. In evaluation, we fully disable expert calls and only measure the standalone policy. The fact that the EAPO-trained policy significantly outperforms Self-Exploratory RL without any expert access indicates that it has indeed absorbed useful reasoning patterns rather than merely outsourcing reasoning to experts.

---

### Official Review · Reviewer_oegD · 2025-10-31

**Soundness:** 2
**Presentation:** 3
**Contribution:** 2
**Rating:** 4
**Confidence:** 3

**Summary:**

This article proposes a novel reinforcement learning framework called EAPO (Expert-Assisted Policy Optimization) that allows policy models to dynamically request the assistance of more powerful expert models during training, integrating expert feedback into their sampling trajectory. The empirical studies show that EAPO consistently outperforms strong baseline methods, including self-exploratory reinforcement learning, expert distillation, and multi-agent collaboration in math problems, demonstrating its effective knowledge transfer and inference capabilities.

**Strengths:**

1. Formulating expert consultation as an optimizable and annealing action in reinforcement learning is a novel idea. It goes beyond static workflows and distillation.
2. Experiments are comprehensive. This paper compares extensive baseline methods including self-exploratory methods and distillation methods on three complex reasoning benchmarks. Results are statistically and qualitatively convincing.
3. The presentation of this paper is clear, with intuitive explanations of complex mechanisms.

**Weaknesses:**

1. EAPO mainly aims to use powerful models for inference in RL processes. In this case, why not directly use large models for inference, which would consume fewer resources.
2. EAPO provides access to significantly stronger expert models. It’s unclear how EAPO performs when the expert is only marginally better or when experts are noisy. An ablation on expert quality would strengthen robustness claims.
3. Parallel expert querying during training increases inference and memory overhead. The paper lacks an analysis of training-time cost against gain trade-offs. The article lacks an analysis of the training process time, such as the inference time for each training step.
4. While promising, all experiments are in mathematical reasoning. Demonstrating EAPO on non-math domains (e.g., code, science) would bolster its generality.
5. EAPO provides a form of intermediate guidance, but the paper does not discuss how this compares to the more established concept of process supervision. (e.g., PRMs)

**Questions:**

1. How does EAPO performance change if the expert model is weaker, for example, the same size as policy or only slightly better? Could EAPO still learn useful strategies for exploration?
2. Do the authors test EAPO on non-math tasks such as HumanEval? Does EAPO generalize to other structured reasoning domains?
3. How does EAPO compare to methods that use learned reward models trained on step-level annotations (e.g., PRMs)?

---

> ### Author Response · Authors · 2025-11-20
> **To Reviewer oegD (1/4)**
>
> Dear Reviewer oegD,
>
> Thank you for your thorough review and constructive feedback. All your concerns have been carefully responded. The manuscript is carefully revised accordingly. We sincerely hope our responses fully address your questions.
>
> **To major concerns:**
>
> **Q1:** *EAPO mainly aims to use powerful models for inference in RL processes. In this case, why not directly use large models for inference, which would consume fewer resources.*
>
> **A1:** Thank you for your comment. In fact, the primary goal of EAPO is not to rely on powerful models to perform inference during RL, but rather to use powerful models as auxiliaries to enhance the independent reasoning ability of the policy model. Our method offers a framework that is different from both standard knowledge distillation and offline-sampling-based RL: the policy model can consult a strong expert model on demand during training, and in doing so, it learns *when* and *how* to query the expert. This allows the policy model to internalize the expert’s knowledge into its own parameters and yields a dynamic and generalizable expert-assisted RL framework.
>
> Therefore, instead of simply depending on a large model at inference time, EAPO aims to train a smaller policy model that, after expert-assisted RL training, can achieve stronger reasoning performance even without access to the expert at test time.
>
> &nbsp;
>
> **Q2:** *EAPO provides access to significantly stronger expert models. It’s unclear how EAPO performs when the expert is only marginally better or when experts are noisy. An ablation on expert quality would strengthen robustness claims. How does EAPO performance change if the expert model is weaker, for example, the same size as policy or only slightly better? Could EAPO still learn useful strategies for exploration?*
>
> **A2:** Thank you for this insightful comment. We add systematic ablations on expert quality to better assess how EAPO behaves when the expert is only slightly stronger or even noisy.
>
> First, we consider the case where the expert has exactly the same size and architecture as the policy. We set the expert identical to the policy and compare EAPO with Self-Exploratory RL. On AIME2024 and AIME2025, EAPO is slightly worse; on AIMO2025, it is slightly better; overall, performance is roughly on par with some fluctuation. Combined with our self-collaboration results, this indicates that when experts and policy have essentially the same capability, EAPO cannot obtain meaningful information gain, and the extra interaction may even amplify redundancy/noise and slightly hurt performance.
>
> Second, we study the case where the expert is only moderately stronger. Here, the policy is Deepseek-R1-distill-Qwen-7B, and the expert is Llama-8B. In this configuration, EAPO outperforms Self-Exploratory RL on AIME2024, AIME2025, and AIMO2025. We attribute this not only to the expert being slightly larger, but also to differing capability distributions: architectural and pretraining differences provide complementary information, which EAPO’s on-demand consultation mechanism can effectively exploit to learn better exploration strategies.
>
> |                     | Expert                 | AIME2024 | AIME2024 | AIME2025 | AIME2025 | AIMO2025 | AIMO2025 | Average | Average |
> | ------------------- | ---------------------- | -------- | -------- | -------- | -------- | -------- | -------- | ------- | ------- |
> |                     |                        | pass@32  | var      | pass@32  | var      | pass@32  | var      | pass@32 | var     |
> | Self-Exploratory RL |                        | 67.16    | 0.0646   | 55.09    | 0.0625   | 55.22    | 0.0911   | 59.16   | 0.0727  |
> | homogeneous EAPO    | Deepseek-R1-distiil-7B | 63.74    | 0.0613   | 54.73    | 0.0570   | 58.08    | 0.1084   | 58.85   | 0.0756  |
> | heterogeneous EAPO  | Llama-8B               | 67.19    | 0.0604   | 55.92    | 0.0651   | 58.90    | 0.0927   | 60.66   | 0.0727  |

---

> ### Author Response · Authors · 2025-11-20
> **To Reviewer oegD (2/4)**
>
> To further evaluate the robustness of EAPO under noisy or low-quality experts, we design two additional experiments:
>
> **High-error experts.** For each problem in AIME2024/2025 and AIMO2025, we build an “error expert” by selecting 3 incorrect trajectories from 32 QwQ-32B samples (if fewer than 3 are incorrect, we sample from all 32). This yields error rates above 50% (53.33% for AIME2024, 63.33% for AIME2025 and AIMO2025). We then evaluate DeepSeek-R1-distill-Qwen-7B on this collaboration dataset and compare it against an Expert-Assisted Workflow built from 3 randomly sampled trajectories.
>
> **Completely meaningless guidance.** In constructing collaboration trajectories, we replace all expert messages with fixed, content-free text (e.g., “I cannot offer you any assistance; please resolve the issue yourself.”), simulating experts that are essentially uninformative.
>
> These settings simulate scenarios where the expert provides only weak or even purely noisy signals, and allow us to examine the stability and failure modes of EAPO under low-quality experts.
>
> |                      | AIME2024 | AIME2024 | AIME2025 | AIME2025 | AIMO2025 | AIMO2025 | Average | Average |
> | -------------------- | -------- | -------- | -------- | -------- | -------- | -------- | ------- | ------- |
> |                      | pass@32  | var      | pass@32  | var      | pass@32  | var      | pass@32 | var     |
> | Random sampling      | 58.07    | 0.2137   | 43.16    | 0.1978   | 46.94    | 0.2003   | 49.39   | 0.2039  |
> | Error sampling       | 53.73    | 0.2025   | 37.54    | 0.2157   | 37.18    | 0.2076   | 42.82   | 0.2086  |
> | meaningless guidance | 49.50    | 0.1906   | 40.05    | 0.2013   | 39.21    | 0.1940   | 42.92   | 0.1953  |
>
> &nbsp;
>
> **Q3:** *Parallel expert querying during training increases inference and memory overhead. The paper lacks an analysis of training-time cost against gain trade-offs. The article lacks an analysis of the training process time, such as the inference time for each training step.*
>
> **A3:** Thank you for your comment. To analyse the trade-off between the additional training-time cost introduced by expert querying and the performance gains, we clarify this from two perspectives:
>
> 1. Training-time cost vs. performance trade-off: Compared to Self-Exploratory RL, EAPO requires roughly 1.4× training time in our experiments. This extra cost mainly comes from invoking the expert model during training to obtain higher-quality trajectories and feedback. However, at test time, we completely disable expert calls, so the inference-time cost of EAPO is the same as that of Self-Exploratory RL.
> 2. Per-step training inference time: In our setup, the training process of EAPO takes about 5.92 minutes per step, using a hardware configuration of 8×8 H20 GPUs.

---

> ### Author Response · Authors · 2025-11-20
> **To Reviewer oegD (3/4)**
>
> **Q4:** *While promising, all experiments are in mathematical reasoning. Demonstrating EAPO on non-math domains (e.g., code, science) would bolster its generality. Do the authors test EAPO on non-math tasks such as* *HumanEval? Does EAPO generalize to other structured reasoning domains?*
>
> **A4:** Thank you for this comment. To better assess the generality of EAPO beyond mathematical reasoning, we have added a series of experiments on non-math benchmarks, including HumanEval, HLE, GPQA, MMLU, EvalPlus, HotpotQA, and simpleQA, which cover code generation, scientific reasoning, and knowledge-intensive multi-step reasoning tasks. Our analysis has two parts:
>
> 1. Expert-assisted workflows on non-math datasets:
>
> We first evaluate self-collaboration and the Expert-Assisted Workflow in an offline setting. We observe that introducing an expert model into the workflow yields consistent and significant performance gains, even for code and science tasks. This indicates that the idea of “expert-assisted reasoning” is not limited to mathematics, but can also extend to other domains that require structured reasoning.
>
> | Method                   | Policy Model | Experts | HumanEval       | HLE                  | GPQA                 | MMLU                 | EvalPlus        | HotpotQA                                 | simpleQA             | Average                    |
> | ------------------------ | ------------ | ------- | --------------- | -------------------- | -------------------- | -------------------- | --------------- | ---------------------------------------- | -------------------- | -------------------------- |
> |                          |              |         | Code Generation | Scientific Reasoning | Scientific Reasoning | Scientific Reasoning | Code Generation | Knowledge-Intensive Multi-Step Reasoning | Scientific Reasoning | Non-Mathematical Benchmark |
> | Base Model               | R1-7B        | --      | 86.79           | 7.00                 | 45.92                | 48.16                | 83.72           | 12.86                                    | 6.70                 | 41.59                      |
> | Base Model               | R1-14B       | --      | 93.90           | 7.47                 | 56.14                | 56.82                | 94.57           | 30.08                                    | 8.95                 | 49.70                      |
> | Base Model               | QwQ-32B      | --      | 59.96           | 8.34                 | 59.67                | 67.03                | 60.61           | 34.09                                    | 12.31                | 43.14                      |
> | Expert-Assisted Workflow | R1-7B        | R1-7B   | 84.76           | 6.69                 | 44.23                | 45.01                | 80.49           | 11.93                                    | 5.74                 | 39.84                      |
> | Expert-Assisted Workflow | R1-7B        | R1-14B  | 83.54           | 7.27                 | 46.06                | 52.59                | 80.91           | 27.85                                    | 8.27                 | 43.78                      |
> | Expert-Assisted Workflow | R1-7B        | QwQ-32B | 66.46           | 7.29                 | 48.18                | 54.96                | 65.15           | 29.45                                    | 11.13                | 40.37                      |
>
> 2. Generalizing EAPO from math-only training to evaluations on non-math tasks:
>
> To directly evaluate the generalization of EAPO, we use only the mathematical-reasoning dataset (dapo-math) for training. We find that, even solely trained  on mathematical reasoning data, EAPO consistently outperforms Self-Exploratory RL on these non-math benchmarks. This demonstrates that expert-assisted policy optimization is a broadly applicable approach rather than one constrained to mathematical reasoning.
>
> |                     | HumanEval | HLE   | GPQA  | MMLU  | EvalPlus | HotpotQA | simpleQA | Average |
> | ------------------- | --------- | ----- | ----- | ----- | -------- | -------- | -------- | ------- |
> | Base Model          | 86.79     | 7.00  | 45.92 | 48.16 | 83.72    | 12.86    | 6.70     | 41.59   |
> | Self-Exploratory RL | 89.34     | 15.96 | 48.13 | 55.47 | 85.16    | 21.46    | 14.56    | 47.15   |
> | EAPO                | 91.56     | 16.07 | 49.92 | 58.61 | 88.04    | 23.75    | 16.82    | 49.25   |
>
> Taken together, these results show that: (i) expert-assisted collaborative workflows remain effective in non-math domains, and (ii) the policy trained with EAPO on math data exhibits transferable and stable gains on other structured reasoning tasks. This provides empirical support for our claim that EAPO generalizes beyond purely mathematical reasoning.

---

> ### Author Response · Authors · 2025-11-20
> **To Reviewer oegD (4/4)**
>
> **Q5:** *EAPO provides a form of intermediate guidance, but the paper does not discuss how this compares to the more established concept of process supervision. (e.g., PRMs) How does EAPO compare to methods that use learned reward models trained on step-level annotations (e.g., PRMs)?*
>
> **A5:** Thank you for this important and highly relevant question. We are happy to clarify how EAPO relates to process supervision methods such as Process Reward Models (PRMs).
>
> 1. No additional reward model or step-level labels PRM-based approaches require collecting step-level annotations and training a separate process reward model, which adds supervision and engineering overhead and ties performance to annotation quality. EAPO does not train any extra reward models and does not rely on step-level labels; we simply indicate in the system prompt that an expert is available, and the policy learns—via RL—*when and how* to consult this expert through natural-language interaction.
> 2. Semantic guidance instead of scalar scores PRMs provide dense but low-information scalar scores for intermediate steps (e.g., correctness or progress). In EAPO, the expert returns natural-language reasoning and suggestions that can explain *why* a step is wrong and *how* to fix or proceed. The policy thus learns from rich, semantic signals rather than reacting only to numeric evaluations.
> 3. Optimizing expert-usage strategy, not a fixed scorer Once trained, a PRM acts as a static external critic whose behavior is fixed by its training data. In EAPO, what we optimize is the policy’s expert-usage strategy—whether to call the expert, what to ask, and how to integrate the feedback—so leveraging the expert becomes part of the policy space itself, rather than a separate scoring module.
> 4. Forward-looking assistance and better generalization PRM signals are primarily backward-looking, evaluating completed trajectories and often needing retraining when moving to new tasks or domains. EAPO instead provides forward-looking, in-context assistance: during rollout, the policy can proactively request help at uncertain points and immediately use the guidance to shape its next actions. Over training, the policy gradually internalizes the expert’s knowledge into its own parameters, so no expert calls are needed at test time.
>
> In this sense, PRM-based process supervision focuses on learning an explicit process reward model to score intermediate steps, whereas EAPO focuses on learning a dynamic, on-demand expert-consultation policy driven by natural-language guidance and embedding that guidance into the model’s exploration and reasoning behavior.

---

### Official Review · Reviewer_EoSa · 2025-11-12

**Soundness:** 3
**Presentation:** 3
**Contribution:** 3
**Rating:** 6
**Confidence:** 2

**Summary:**

This paper proposes a reinforcement learning framework called Expert-Assisted Policy Optimization (EAPO), which integrates external expert models (e.g., larger LLMs) during policy training. The goal is to leverage expert feedback to provide richer rewards, accelerate convergence, and improve learning stability compared to standard RL optimization.

**Strengths:**

1. Paper is well written and motivation is clearly stated.
2. Experiments support the effectiveness of their proposed RL framework.

**Weaknesses:**

1. The "expert" idea is not novel. Many prior works have explored using larger or teacher models to guide smaller student models. Similar approaches have also been studied, such as using supervised data (human or model-generated) to improve RL. This paper lacks a clear discussion and comparison with those related methods.
2. The paper does not explicitly state: how the framework decides whether experts is required at time t; how to determine which instances are simple, hard, or complex at the beginning of training
3. Notation typos. The subscript in Line 121-123 is not consistent with Line120; Line 124 should use index t-1 based on the equation in Line 126; the meaning of q_t in Line 235 is unclear.

**Questions:**

1. What is the Expert-Assisted Workflow method mentioned, and how does it differ from your proposed EAPO?
2. Have you considered the compute cost of using experts? Calling larger models can be expensive, and standard RL might generate more trajectories within the same compute budget.
3. It seems the main benefit of using experts is to make the initial reward signals denser so that more meaningful trajectories are used for model updates. Could you show how the initial rewards change after introducing experts?

---

> ### Author Response · Authors · 2025-11-20
> **To Reviewer EoSa (1/3)**
>
> Dear Reviewer EoSa,
>
> Thank you for your thorough review and constructive feedback. All your concerns have been carefully responded. The manuscript is carefully revised accordingly. We sincerely hope our responses fully address your questions.
>
> **To major concerns:**
>
> **Q1:** *The "expert" idea is not novel. Many prior works have explored using larger or teacher models to guide smaller student models. Similar approaches have also been studied, such as using supervised data (human or model-generated) to improve* *RL. This paper lacks a clear discussion and comparison with those related methods.*
>
> **A1:** Thank you for your thoughtful comment. We argue that our work is substantially different from prior research in both mechanism and application paradigm. Existing methods that use larger/teacher models or supervised data to aid RL are mostly static: teacher data must be pre-collected or curated and then injected into training at fixed stages. In contrast, our method lets the policy learn during RL training when and how to consult an expert, forming a dynamic, on-demand consultation mechanism. The model is no longer tied to pre-prepared supervision for each task, but can query experts as needed and internalize their feedback to continually improve. Empirically, our approach consistently outperforms standard distillation and static supervision-based baselines on AIME2024, AIME2025, and AIMO2025, providing concrete evidence that it goes beyond existing “teacher-guided” approaches in both novelty and effectiveness.
>
> &nbsp;
>
> **Q2:** *The paper does not explicitly state: how the framework decides whether experts is required at time t; how to determine which instances are simple, hard, or complex at the beginning of training.*
>
> **A2:** Thank you for your comment. One of the key goals of the EAPO framework is precisely to enable the model to learn when and how to consult experts during training. Consequently, whether an expert is required at time step t is not determined by any hand-crafted rule; instead, it is chosen by the model itself based on its current state.
>
> Likewise, our method does not require us to pre-annotate training instances as “simple”, “hard”, or “complex” before training. Rather, through interaction and optimization, the model gradually learns to infer the difficulty of a problem and to decide whether, and in what manner, to invoke experts. Empirically, we observe three typical behaviors: for relatively simple instances, the model tends to solve the problem entirely on its own; for harder or more complex cases, it is more likely to rely on experts.
>
> Crucially, these behaviors are learned during training, not imposed through any hard-coded rules. In the revised version of the paper, shown in Section 2.3, we will more clearly explain (i) how the framework decides whether experts are invoked at time t, and (ii) the observation of “simple / hard / complex” instances.
>
> &nbsp;
>
> **Q3:** *Notation typos. The subscript in Line 121-123 is not consistent with Line120; Line 124 should use index t-1 based on the equation in Line 126; the meaning of q_t in Line 235 is unclear.*
>
> **A3:** Thank you for your comment. The inconsistency in notation has been corrected in the latest version of the paper.

---

> ### Author Response · Authors · 2025-11-20
> **To Reviewer EoSa (2/3)**
>
> **Q4:** *What is the Expert-Assisted* *Workflow* *method mentioned, and how does it differ from your proposed EAPO?*
>
> **A4:** Thank you for your comment. The Expert-Assisted Workflow in our paper is an offline, non-training, inference-only workflow. For each problem in AIME2024, AIME2025, and AIMO2025, we sample 96 independent reasoning trajectories from QwQ-32B. Every 3 trajectories are concatenated using our prompt template to form 32 collaboration trajectories per problem, which are then fed into Deepseek-R1-distill-Qwen-7B to produce the final answer. The choice of 96 samples aligns with the pass@32 setting for fair comparison with test-time scaling baselines and to stabilize results. This workflow is entirely static and hand-crafted: expert usage is fully pre-defined, and the student’s parameters are never updated.
>
> By contrast, EAPO is an RL-based training framework. We do not specify when to call experts or what to ask; we only indicate in the system prompt that an expert is available. During training, the policy learns a dynamic expert-calling strategy, when and how to consult experts, and gradually internalizes their knowledge into its own parameters. At test time, the model achieves strong reasoning performance without any expert access.
>
> ```Python
> PROMPT_TEMPLATE = """Problem: {question}
> You have been provided with a set of responses from various open-source agents to the latest user query. Your task is to synthesize these responses into a single, high-quality response. It is crucial to critically evaluate the information provided in these responses, recognizing that some of it may be biased or incorrect. Your response should not simply replicate the given answers but should offer a refined, accurate, and comprehensive reply to the instruction. Ensure your response is well-structured, coherent, and adheres to the highest standards of accuracy and reliability.
> Responses from agents:
> Expert A: {expert_a}
> Expert B: {expert_b}
> Expert C: {expert_c}"""
> ```
>
> &nbsp;
>
> **Q5:** *Have you considered the compute cost of using experts? Calling larger models can be expensive, and standard RL might generate more trajectories within the same compute budget.*
>
> **A5:** Thank you for your comment. We have carefully considered the computational cost of using experts in EAPO, and our design mitigates it in several ways. First, EAPO uses on-demand expert consultation: whether to call an expert is decided by the policy itself, rather than being enforced for every step or sample, so the actual expert use rate is much lower than in methods that rely on large-scale supervised data . Second, we employ an annealing scheme that gradually reduces expert usage over training, which both lowers total compute and encourages the model to rely increasingly on its own reasoning. Third, at test time, we disable expert calls entirely and evaluate only the trained policy, so expert computing is incurred only during training and does not affect deployment-time costs.
>
> Regarding the point that standard RL can generate more trajectories under the same compute budget, our experiments suggest that simply increasing the number of trajectories does not guarantee better performance and can even be harmful due to redundancy and noise (as seen in our self-collaboration results in Table 1). In contrast, using a stronger expert to produce fewer but higher-quality, information-dense trajectories during training is often more effective than scaling up low-quality trajectories under the same compute budget.

---

> ### Author Response · Authors · 2025-11-20
> **To Reviewer EoSa (3/3)**
>
> **Q6:** *It seems the main benefit of using experts is to make the initial reward signals denser so that more meaningful trajectories are used for model updates. Could you show how the initial rewards change after introducing experts?*
>
> **A6:** Thank you for your comment. To directly address your question about how the initial rewards change after introducing experts, we conducted a detailed analysis of the early-stage reward signals, focusing on:
>
>  (1) the mean of the critic score (Mean) over the first 5–40 training steps;
>
>  (2) the proportion of low-magnitude rewards under different thresholds (score ≤ 0.2 / 0.3 / 0.4).
>
> | Method              | Steps≤ | Mean  | score≤0.2 | score≤0.3 | score≤0.4 |
> | ------------------- | ------ | ----- | --------- | --------- | --------- |
> | Self-Exploratory RL | 5      | 0.000 | 70%       | 98%       | 100%      |
> | EAPO                | 5      | 0.132 | 43%       | 63%       | 75%       |
> | Self-Exploratory RL | 10     | 0.068 | 77%       | 97%       | 100%      |
> | EAPO                | 10     | 0.234 | 40%       | 50%       | 67%       |
> | Self-Exploratory RL | 15     | 0.138 | 75%       | 95%       | 100%      |
> | EAPO                | 15     | 0.378 | 36%       | 28%       | 50%       |
> | Self-Exploratory RL | 20     | 0.179 | 67%       | 93%       | 100%      |
> | EAPO                | 20     | 0.427 | 22%       | 21%       | 33%       |
> | Self-Exploratory RL | 30     | 0.175 | 70%       | 90%       | 100%      |
> | EAPO                | 30     | 0.461 | 17%       | 16%       | 34%       |
> | Self-Exploratory RL | 40     | 0.196 | 60%       | 88%       | 98%       |
> | EAPO                | 40     | 0.452 | 10%       | 9%        | 28%       |
>
> The results show that, with experts, the reward signals become denser  from the beginning of training:
>
> - Mean: EAPO's average reward is approximately 2-3 times higher than the baseline (without expert), indicating a stronger gradient signal in the early stages.
> - Low Reward Ratio: In the 5-40 step range, almost 70-100% of batches in the baseline fall within the score range ≤ 0.2 / 0.3 / 0.4, while EAPO consistently shows a lower proportion of low rewards compared to the baseline.
> - Overall Trend: The comparison of the mean and the low reward ratio together supports the conclusion that experts make the initial reward signal denser and more informative.

---

### Official Review · Reviewer_CHBo · 2025-11-12

**Soundness:** 2
**Presentation:** 2
**Contribution:** 2
**Rating:** 2
**Confidence:** 4

**Summary:**

This paper proposes Expert-Assisted Policy Optimization (EAPO), a framework that integrates multi-turn interactions with external experts throughout the training process. Unlike prior methods where policy models reason in isolation, EAPO enables the policy to adaptively determine when and how to consult experts, thus obtaining richer reward signals and more reliable reasoning trajectories. Experiments on mathematical reasoning benchmarks like AIME 2024, AIME 2025, and AIMO 2025 demonstrate that EAPO outperforms baselines.

**Strengths:**

1. The paper is well-organized and easy to follow, and the figures are clear and helpful.

2. The paper’s ablation studies, focusing on expert parallelism and expert model size, validate EAPO’s key designs: parallel expert queries outperform sequential ones and self-exploratory RL, while larger experts (e.g., QwQ-32B) outperform smaller ones.

**Weaknesses:**

1. This paper only validates EAPO’s performance on mathematical reasoning benchmarks, including AIME and AIMO, while excluding other domains’ datasets like HLE and GPQA.

2. The expert model-related overhead is relatively high. EAPO adopts a multi-expert parallel reasoning mechanism to improve information coverage and relies on large-scale expert models to ensure the quality of guidance, which undoubtedly increases the consumption of computing resources during model training and deployment, limiting its practicality in resource-constrained scenarios.

3. EAPO is highly dependent on the capabilities of expert models. Experiments show that using smaller-scale expert models will lead to a significant decline in EAPO's performance. Moreover, when the expert is just a single model or is identical to the policy model, it may be difficult to effectively assist policy model optimization due to the lack of diverse perspectives.

4. In terms of feedback mechanism design, EAPO is similar in idea to related works that rely on reward models to provide feedback. It does not form a significant difference in core feedback logic or framework design, resulting in insufficient novelty compared with existing reinforcement learning-assisted optimization methods.

5. Minor issues in Section 2: In Line 122, how to generate the expert assistance $o_i$
is unclear, in Line 126, $\pi_\theta(\tau_t, \alpha_t | H_{t-1}) = \pi_\theta^\tau(\tau_t | H_{t-1}) \cdot \pi_\theta^\alpha(\alpha_t | H_{t-1}, \tau_t)$ is problematic, since the formula is given $H_{t-1}$ instead of $H_t$, and in Line235, the $q_i$ is undefined.

**Questions:**

1. How does EAPO perform on other domains’ datasets like HLE and GPQA?

2. How robust is EAPO to low-quality or noisy expert models? For example, when experts provide incorrect or meaningless guidance, does the framework’s performance degrade gracefully? And how could you solve this problem?

3. How dependent is EAPO on the expert models? What happens if the experts are fewer, weaker, or partially misaligned? Specifically, if the expert model is identical to the policy model, how do performance and stability change?

---

> ### Author Response · Authors · 2025-11-20
> **To Reviewer CHBo (1/6)**
>
> Dear Reviewer CHBo,
>
> Thank you for your thorough review and constructive feedback. All your concerns have been carefully responded as follows. The manuscript is carefully revised accordingly. We sincerely hope our responses fully address your questions.
>
> **To major concerns:**
>
> **Q1:** *This paper only validates EAPO’s performance on mathematical reasoning benchmarks, including* *AIME* *and AIMO, while excluding other domains’ datasets like HLE and* *GPQA. How does EAPO perform on other domains’ datasets like HLE and GPQA?*
>
> **A1:** Thank you for this comment. To better assess the generality of EAPO beyond mathematical reasoning, we have added a series of experiments on non-math benchmarks, including HumanEval, HLE, GPQA, MMLU, EvalPlus, HotpotQA, and simpleQA, which cover code generation, scientific reasoning, and knowledge-intensive multi-step reasoning tasks. Our analysis has two parts:
>
> 1. Expert-assisted workflows on non-math datasets:
>
> We first evaluate self-collaboration and the Expert-Assisted Workflow in an offline setting. We observe that introducing an expert model into the workflow yields consistent and significant performance gains, even for code and science tasks. This indicates that the idea of “expert-assisted reasoning” is not limited to mathematics, but can also extend to other domains that require structured reasoning.
>
> | Method                   | Policy Model | Experts | HumanEval       | HLE                  | GPQA                 | MMLU                 | EvalPlus        | HotpotQA                                 | simpleQA             | Average                    |
> | -------------- | ------------ | ------- | --------------- | -------- | -------------------- | -------------------- | --------------- | ---------------------------------------- | --------------- | ------------------ |
> |                          |              |         | Code Generation | Scientific Reasoning | Scientific Reasoning | Scientific Reasoning | Code Generation | Knowledge-Intensive Multi-Step Reasoning | Scientific Reasoning | Non-Mathematical Benchmark |
> | Base Model               | R1-7B        | --      | 86.79           | 7.00                 | 45.92                | 48.16                | 83.72           | 12.86                                    | 6.70                 | 41.59                      |
> | Base Model               | R1-14B       | --      | 93.90           | 7.47                 | 56.14                | 56.82                | 94.57           | 30.08                                    | 8.95                 | 49.70                      |
> | Base Model               | QwQ-32B      | --      | 59.96           | 8.34                 | 59.67                | 67.03                | 60.61           | 34.09                                    | 12.31                | 43.14                      |
> | Expert-Assisted Workflow | R1-7B        | R1-7B   | 84.76           | 6.69                 | 44.23                | 45.01                | 80.49           | 11.93                                    | 5.74                 | 39.84                      |
> | Expert-Assisted Workflow | R1-7B        | R1-14B  | 83.54           | 7.27                 | 46.06                | 52.59                | 80.91           | 27.85                                    | 8.27                 | 43.78                      |
> | Expert-Assisted Workflow | R1-7B        | QwQ-32B | 66.46           | 7.29                 | 48.18                | 54.96                | 65.15           | 29.45                                    | 11.13                | 40.37                      |
>
>
> 2. Generalizing EAPO from math-only training to evaluations on non-math tasks:
>
> To directly evaluate the generalization of EAPO, we use only the mathematical-reasoning dataset (dapo-math) for training. We find that, even solely trained  on mathematical reasoning data, EAPO consistently outperforms Self-Exploratory RL on these non-math benchmarks. This demonstrates that expert-assisted policy optimization is a broadly applicable approach rather than one constrained to mathematical reasoning.
>
> |                     | HumanEval | HLE   | GPQA  | MMLU  | EvalPlus | HotpotQA | simpleQA | Average |
> | ------------------- | --------- | ----- | ----- | ----- | -------- | -------- | -------- | ------- |
> | Base Model          | 86.79     | 7.00  | 45.92 | 48.16 | 83.72    | 12.86    | 6.70     | 41.59   |
> | Self-Exploratory RL | 89.34     | 15.96 | 48.13 | 55.47 | 85.16    | 21.46    | 14.56    | 47.15   |
> | EAPO                | 91.56     | 16.07 | 49.92 | 58.61 | 88.04    | 23.75    | 16.82    | 49.25   |
>
> Taken together, these results demonstrate that: (i) expert-assisted workflows remain effective in non-math domains, and (ii) the policy model trained with EAPO on math data exhibits stable and transferable improvements on other structured reasoning tasks. These findings provide empirical support for the claim that EAPO generalizes beyond purely mathematical reasoning.

---

> ### Author Response · Authors · 2025-11-20
> **To Reviewer CHBo (2/6)**
>
> **Q2:** *The expert model-related overhead is relatively high. EAPO adopts a multi-expert parallel reasoning mechanism to improve information coverage and relies on large-scale expert models to ensure the quality of guidance, which undoubtedly increases the consumption of computing resources during model training and deployment, limiting its practicality in resource-constrained scenarios.*
>
> **A2:** Thank you for your comment. We address the concern about expert-related overhead and practicality in resource-constrained settings as follows:
>
> First, regarding training-time cost, EAPO requires approximately 1.4× the training time of self-exploratory RL in our experiments. This overhead mainly comes from on-demand consultants, which are used to obtain higher-quality trajectories and intermediate guidance. However, these consultants are fully disabled at test time, making the inference-time cost of EAPO identical to that of self-exploratory RL.
>
> Second, for resource-constrained scenarios, we experimented with sequential expert querying . Our results show that, even with purely sequential expert calls, EAPO still yields clear improvements over Self-Exploratory RL. What's more, leveraging the parallel inference capabilities provided by vLLM servers, even in scenarios with limited computing resources, a single model can be executed in parallel, achieving efficient resource reuse and amplifying business capabilities. Furthermore, our expert models also support invocation via OpenAI-compatible APIs. Therefore, EAPO is not limited by specific computing power scale or deployment form, but rather is a general policy optimization strategy that can be flexibly embedded into different systems and architectures.
>
> Third, compared with approaches that depend on a separately trained reward model for process supervision, EAPO uses a dynamically on-demand expert-calling mechanism instead of consulting experts at every step, which helps avoid unnecessary queries and reduces wasted compute. In addition, we apply an annealing schedule in the later stages of training to gradually reduce the expert call probability, pushing the policy to rely more on its own reasoning. As a result, both the expert call rate and theassociated compute cost decrease significantly over the course of training.

---

> ### Author Response · Authors · 2025-11-20
> **To Reviewer CHBo (3/6)**
>
> **Q3:** *EAPO is highly dependent on the capabilities of expert models. Experiments show that using smaller-scale expert models will lead to a significant decline in EAPO's performance. Moreover, when the expert is just a single model or is identical to the policy model, it may be difficult to effectively assist policy model optimization due to the lack of diverse perspectives. How dependent is EAPO on the expert models? What happens if the experts are fewer, weaker, or partially misaligned? Specifically, if the expert model is identical to the policy model, how do performance and stability change? How robust is EAPO to low-quality or noisy expert models? For example, when experts provide incorrect or meaningless guidance, does the framework’s performance degrade gracefully? And how could you solve this problem?*
>
> **A3:** Thank you for your comprehensive and insightful comment. We will now answer your questions in more detail in four parts, including (1) analysis of homogeneous and heterogeneous experts, (2) analysis of serial and parallel consultation experts, (3) analysis of conflicting opinions among experts, and (4) robustness analysis of low-quality and noisy expert opinions.
>
> | index |                     | Expert                 | AIME2024 | AIME2024 | AIME2025 | AIME2025 | AIMO2025 | AIMO2025 | Average | Average |
> | ----- | ------------------- | ---------------------- | -------- | -------- | -------- | -------- | -------- | -------- | ------- | ------- |
> |       |                     |                        | pass@32  | var      | pass@32  | var      | pass@32  | var      | pass@32 | var     |
> | (1)   | Self-Exploratory RL |                        | 67.16    | 0.0646   | 55.09    | 0.0625   | 55.22    | 0.0911   | 59.16   | 0.0727  |
> | (2)   | homogeneous EAPO    | Deepseek-R1-distiil-7B | 63.74    | 0.0613   | 54.73    | 0.0570   | 58.08    | 0.1084   | 58.85   | 0.0756  |
> | (3)   | heterogeneous EAPO  | Llama-8B               | 67.19    | 0.0604   | 55.92    | 0.0651   | 58.90    | 0.0927   | 60.66   | 0.0727  |
>
> (1) Dependence on expert strength: homogeneous vs. heterogeneous experts
>
> We first examine the behavior of EAPO when the expert is identical to or only slightly stronger than the policy:
>
> **Expert** **identical to the policy (homogeneous expert pool)**, which is shown in row (2): When the expert is identical to the policy, EAPO performs roughly on par with Self-Exploratory RL: slightly worse on AIME2024/2025, slightly better on AIMO2025, without collapse. Combined with our self-collaboration analysis, this supports that highly overlapping experts provide minimal information gain, and redundant signals may introduce mild noise.
>
> **Slightly stronger and heterogeneous expert**, which is shown in row (3): With DeepSeek-R1-distill-Qwen-7B as policy and Llama-8B as expert, EAPO consistently improves over Self-Exploratory RL across all benchmarks. The gains come not only from scale but also from complementary inductive biases. EAPO leverages on-demand consulting to exploit this complementarity.
>
> (2) Fewer experts: single-expert / sequential querying
>
> Our ablations already evaluate sequential querying (effectively K=1), as reported in Table 2 in the paper. Results show: Sequential querying still outperforms Self-Exploratory RL, demonstrating that even a single expert brings information gain. But sequential querying is worse than parallel querying, indicating that multiple experts and diverse viewpoints provide richer coverage and better performance. Thus, reducing expert count lowers gains but does not cause instability, and EAPO remains better than Self-Exploratory RL.
>
> We also clarify the motivation: EAPO injects rich, heterogeneous expert signals directly into RL. As long as the expert’s capability distribution differs from the policy, EAPO can exploit this complementarity. The novelty lies in integrating expert consultation early in RL, so the policy internalizes expert knowledge and requires no experts at test time.

---

> ### Author Response · Authors · 2025-11-20
> **To Reviewer CHBo (4/6)**
>
> (3) Conflicting opinions among experts
>
> In our offline workflow experiments, we observe cases where all experts produce different but incorrect final answers; yet the model can still read their reasoning chains, compare them, spot the flaws, and derive the correct solution. This shows that in EAPO or workflow-style setups, the model is not simply voting over expert outputs, but mainly exploiting their natural-language reasoning as signals. Even when experts have conflicting wrong opinions, their reasoning traces remain useful, which underscores a key advantage of EAPO over scalar process supervision: language-based guidance provides a much richer structure than a single process score.
>
> ```Plain
> System Prompt:
>
> Role Declaration
> You function as a collaborative hub agent, forming an equal partnership with Experts A/B/C/D in a cooperative network. You are responsible for integrating your own knowledge with that of the other experts. Your tasks include:  \n1. Scanning knowledge anchors  \n2. Verifying and supplementing knowledge to refine the reasoning process  \n3. Executing the final decision and providing the answer to the question  \n\n
>
> Input Data
> Question: Consider a process which outputs a random English letter with uniform probability (i.e., each with probability 1/26). What is expected time until a sequence \"TENETENET\" appears?  \n
>
> Expert Proposals
> Expert A: Exact Answer: 5,429,503,678,976 letters (approximately 5.43 trillion letters)\n\nConfidence: 95%\n\nI'm highly confident in this approach as it's a well-established principle in probability theory. The only minor uncertainty comes from potential edge cases or overlapping patterns, but in this case, they don't affect the fundamental calculation.  \n
> Expert B: Exact Answer: \\(542951556296\\)\nConfidence: \\(100\\%\\)  \n
> Expert C: Exact Answer: \\(26^9\\)\nConfidence: \\(95\\%\\)  \n
> Expert D: Exact Answer: Approximately \\( 3195168 \\) steps.\n\nConfidence: 90%  \n\n
>
> ======================================================================================
>
> Response:
>
> Explanation: After analyzing all expert inputs, I conclude that this problem requires careful consideration of pattern overlaps in a sequence of independent random letters. Experts A and C suggest a simple approach using 26^9, while Expert B provides the most rigorous analysis by considering self-overlaps of the pattern \"TENETENET\". The pattern has self-overlaps of lengths 5 and 1, which affects the expected waiting time. Expert B's approach using the formula E(X)=26^9 + 26^5 + 26^1 is the most accurate as it accounts for these overlaps. Expert D's answer appears to be incorrect by several orders of magnitude. The correct calculation must account for: 1) The base probability of the full sequence (26^9), 2) The probability contributions from self-overlaps of length 5 (\"TENET\") and length 1 (\"T\"). Using the self-overlap principle and the sum 26^9 + 26^5 + 26^1, we get 5429503678976 + 11881376 + 26.
>
> Exact Answer: 5429515560378 ✅
> Confidence: 98%
>
> ======================================================================================
>
> Ground Truth: 5429515560378
> ```

---

> ### Author Response · Authors · 2025-11-20
> **To Reviewer CHBo (5/6)**
>
> (4) Robustness to low-quality and noisy experts
>
> To directly test robustness under low-quality or noisy experts, we design two extreme settings:
>
> **Low-quality experts.** For each problem in AIME2024/2025 and AIMO2025, we build an “error expert” by selecting 3 incorrect trajectories from 32 QwQ-32B samples (if fewer than 3 are incorrect, we sample from all 32 trajectories). This yields error rates above 50% (53.33% for AIME2024, 63.33% for AIME2025 and AIMO2025). We then evaluate DeepSeek-R1-distill-Qwen-7B on this collaboration dataset and compare it against an Expert-Assisted Workflow built from 3 randomly sampled trajectories.
>
> **Completely meaningless guidance.** In constructing collaboration trajectories, we replace all expert messages with fixed, content-free text (e.g., “I cannot offer you any assistance; please resolve the issue yourself.”), simulating experts that are essentially uninformative.
>
> |                      | AIME2024 | AIME2024 | AIME2025 | AIME2025 | AIMO2025 | AIMO2025 | Average | Average |
> | -------------------- | -------- | -------- | -------- | -------- | -------- | -------- | ------- | ------- |
> |                      | pass@32  | var      | pass@32  | var      | pass@32  | var      | pass@32 | var     |
> | Random sampling      | 58.07    | 0.2137   | 43.16    | 0.1978   | 46.94    | 0.2003   | 49.39   | 0.2039  |
> | Error sampling       | 53.73    | 0.2025   | 37.54    | 0.2157   | 37.18    | 0.2076   | 42.82   | 0.2086  |
> | meaningless guidance | 49.50    | 0.1906   | 40.05    | 0.2013   | 39.21    | 0.1940   | 42.92   | 0.1953  |
>
> Across both settings, performance degrades as expert quality decreases, but the degradation is smooth and controlled rather than catastrophic: we do not observe RLHF-style “reward hacking,” where the policy exploits systematic biases in expert feedback. As with supervised distillation or standard supervised fine-tuning, a large amount of low-quality data can hurt performance, and EAPO is not exempt from this general limitation. However, EAPO offers a practical mitigation: experts can be filtered in a preprocessing stage by measuring their capability distribution relative to the policy and keeping only those likely to provide positive information gain, while excluding clearly weaker or overly similar experts. This pre-selection reduces the impact of low-quality experts at the source and improves robustness to noisy guidance.

---

> ### Author Response · Authors · 2025-11-20
> **To Reviewer CHBo (6/6)**
>
> **Q4:** *In terms of feedback mechanism design, EAPO is similar in idea to related works that rely on reward models to provide feedback. It does not form a significant difference in core feedback logic or framework design, resulting in insufficient novelty compared with existing reinforcement learning-assisted optimization methods.*
>
> **A4:** Thank you for the comment, but we respectfully disagree that EAPO is not meaningfully different from reward-model–based feedback. EAPO diverges from both outcome-reward and process-reward paradigms in several fundamental aspects:
>
> 1. No Additional Reward Model PRM-based approaches require collecting step-level annotations and training a separate process reward model, adding data and system overhead. EAPO does not train any reward model and does not rely on step-level labels; the policy simply learns, through RL, when and how to consult an expert via natural-language interactions.
> 2. Semantic Guidance Instead of Scalar Scores PRMs provide dense but low-information scalar scores. EAPO uses the expert’s natural-language reasoning and suggestions, which offer actionable explanations about *why* a step is wrong and *how* to fix it. The model learns to interpret these semantic signals rather than react to numeric evaluations.
> 3. Optimizing Expert-Usage Strategy, Not a Fixed Scorer PRMs are fixed once trained. In EAPO, what is optimized is the policy’s *strategy* for invoking and using expert assistance. Leveraging the expert becomes part of the policy space itself, rather than a static external critic.
> 4. Forward-Looking Assistance, Not Retrospective Judgment PRM signals are backward-looking—they evaluate completed trajectories. EAPO provides forward-looking, in-context assistance: the policy can proactively request help at uncertain points during rollout and use the returned guidance to shape immediate next actions.
>
> In short, across the source of feedback (reward model vs. on-demand expert), form of feedback (scalar vs. natural language), optimization objective (reward model vs. assistance strategy), and temporal role of feedback (judgment vs. proactive guidance), EAPO fundamentally differs from standard reward-model–based methods.
>
> &nbsp;
>
> **Q5:** *Minor issues in Section 2: In Line 122, how to generate the expert assistance  is unclear, in Line 126, $\pi_\theta(\tau_t,\alpha_t\mid H_{t-1})=\pi_\theta^\tau(\tau_t \mid H_{t-1})
> \cdot \pi_\theta^{{\alpha}}(\alpha_t \mid H_{t-1},\tau_t)$ is problematic, since the formula is given $H_{t-1}$ instead of $H_t$, and in Line235, the $q_i$ is undefined.*
>
> **A5:** Thank you for your comments. The issues of unclear content and missing definitions have been corrected in the latest version of the paper. Regarding your question on how the expert-assistance queries are generated: in our setup, the policy model first produces a natural-language question it wants to consult. We then directly send this policy-generated question to the expert model’s vLLM server for decoding, and use the decoded response as the expert assistance signal.

---

### Author Response · Authors · 2025-11-25
**Global Response (1/2)**

Dear PCs, SACs, ACs, and Reviewers,

Thank you very much for your valuable contributions to our work. And we sincerely thank all reviewers for their time, feedback, and suggestions. To assist the newly assigned AC and help reduce their workload, we provide below a summary of the key points from the reviews and the rebuttals.

**Strength.** We thank the reviewers for their valuable feedback and positive assessments. The main strengths of the paper, as consistently noted by the reviewers, can be summarized in the following three points:

- **[A novel policy optimization framework]** **Expert-Assisted Policy Optimization (EAPO)** targets the challenge that RL-trained LLMs under verifiable rewards still suffer from inefficient exploration and sparse rewards. EAPO is an RL framework that lets a policy model decide when and how to consult external experts during training, then internalizes their feedback, so the final policy solves problems independently at test time with more reliable reasoning trajectories *(oegD: Strength 1; Ka6W: Strength 1)*.
- **[Strong Generalization and Robustness Across Multiple Reasoning Domains]** We evaluated EAPO **across 10 benchmarks** spanning mathematical reasoning, code generation, scientific reasoning, and knowledge-intensive multi-step reasoning. Overall, EAPO delivers strong and consistent gains: on average, it scores **8 points higher than the base model** and **3 points higher than self-exploration RL**. Breaking this down further, EAPO improves over self-exploration RL by approximately **5 points** on three mathematical reasoning datasets, and by about **2 points** on seven non-mathematical reasoning benchmarks. These comprehensive results demonstrate that EAPO serves as an **effective and general policy optimization framework**, enabling LLMs to dynamically decide when and how to consult experts, internalize expert guidance, and ultimately achieve more robust and generalizable reasoning across diverse domains *(CHBo: Strength 2; EoSa: Strength 2; oegD: Strength 2; Ka6W: Strength 2)*.
- **[High Readability and Full Reproducibility]** The paper is well-structured, clearly motivated, and supported by informative figures and tables, a point unanimously recognized by all 4 reviewers. It provides intuitive explanations for complex mechanisms *(CHBo: Strength 1; EoSa: Strength 1; oegD: Strength 3)*. The work is highly reproducible, offering detailed descriptions of experimental settings in Section 3.1, hyperparameters in Appendix A, prompts in Appendix C, and model deployment configurations in Appendix D *(Ka6W: Strength 3)*.

---

> ### Author Response · Authors · 2025-12-02
> **Global Response (2/2)**
>
> **Concerns and Our Addressing.** During the discussion period, we actively addressed the reviewers' concerns. Although we have not yet received a response from the reviewers, we believe we have resolved all concerns raised by them. Specifically:
>
> - Generalizability of the EAPO framework. The initial submission evaluated EAPO only on mathematical reasoning, raising concerns about its applicability to other domains. *(CHBo: Weakness 1 & Question 1; oegD: Weakness 4 & Question 2; Ka6W: Weakness 1 & Question 1)* We have added broad generalization experiments, covering **7 additional datasets** in code generation, scientific reasoning, and knowledge-intensive multi-step reasoning (HumanEval, HLE, GPQA, MMLU, EvalPlus, HotpotQA, SimpleQA). The results show that EAPO outperforms the **base model by +8 points on average** across **10 benchmarks**, and surpasses **self-exploration RL by +3 points on average**, which demonstrate that EAPO exhibits strong cross-domain generalization and can serve as a universal framework for policy optimization.
>
> |      | AIME2024 | AIME2025 | AIMO2025 | HumanEval | EvalPlus | HLE   | GPQA  | MMLU  | simpleQA | HotpotQA | Average |
> |-|-|-|-|-|-|-|-|-|-|-|-|
> | Base Model   | 51.73    | 37.19    | 38.68    | 86.79     | 83.72    | 7.00  | 45.92 | 48.16 | 6.70     | 12.86    | 41.88   |
> | Self-Exploratory RL | 67.16    | 55.09    | 55.22    | 89.34     | 85.16    | 15.96 | 48.13 | 55.47 | 14.56    | 21.46    | 50.76   |
> | EAPO (Ours)   | 70.03    | 58.01    | 64.17    | 91.56     | 88.04    | 16.07 | 49.92 | 58.61 | 16.82    | 23.75    | 53.70   |
>
> - Ablation studies on expert quality and robustness. Reviewers raised concerns about potential performance degradation when using smaller experts, robustness under noisy or low-quality expert feedback, and comparisons between homogeneous and heterogeneous experts. *(CHBo: Weakness 3 & Question 2,3; oegD: Weakness 2 & Question 1; Ka6W: Weakness 2 & Question 2)* We added two sets of ablations examining (i) homogeneous vs. heterogeneous small-scale experts. Results show that heterogeneous experts can bring an **average improvement of 2 points** compared to homogeneous experts. This indicates that even with small experts, EAPO can exploit heterogeneity to achieve complementary strengths between policy and experts, leading to more effective optimization.
>
> |    | Expert   | AIME2024 | AIME2024 | AIME2025 | AIME2025 | AIMO2025 | AIMO2025 | Average | Average |
> | --- | -- | -------- | --- | -- | ---| --- | -------- | ------- | ------- |
> |      |      | pass@32 $\uparrow$  | var   $\downarrow$   | pass@32 $\uparrow$  | var   $\downarrow$   | pass@32 $\uparrow$  | var    $\downarrow$  | pass@32 $\uparrow$ | var  $\downarrow$   |
> | Self-Exploratory RL |    | 67.16    | 0.0646   | 55.09    | 0.0625   | 55.22    | 0.0911   | 59.16   | 0.0727  |
> | homogeneous EAPO    | R1-7B    | 63.74    | 0.0613   | 54.73    | 0.0570   | 58.08    | 0.1084   | 58.85   | 0.0756  |
> | heterogeneous EAPO  | Llama-8B | 67.19    | 0.0604   | 55.92    | 0.0651   | 58.90    | 0.0927   | 60.66   | 0.0727  |
>
> - (ii) Noisy or low-quality expert responses. The results show that the randomly sampled expert responses we used performed best, scoring an average of **7 points higher** than the other two approaches.
>
> |  | AIME2024 | AIME2024 | AIME2025 | AIME2025 | AIMO2025 | AIMO2025 | Average | Average |
> |-|-|-|-|-|-|-|-|-|
> |    | pass@32 $\uparrow$ | var   $\downarrow$   | pass@32 $\uparrow$ | var $\downarrow$ | pass@32 $\uparrow$ | var $\downarrow$ | pass@32 $\uparrow$ | var $\downarrow$ |
> | Error sampling   | 53.73    | 0.2025   | 37.54    | 0.2157   | 37.18    | 0.2076   | 42.82   | 0.2086  |
> | meaningless guidance | 49.50    | 0.1906   | 40.05    | 0.2013 | 39.21    | 0.1940   | 42.92   | 0.1953  |
> | Random sampling (Settings in EAPO) | 58.07  | 0.2137 | 43.16  | 0.1978   | 46.94    | 0.2003   | 49.39   | 0.2039  |
>
> - Expert cost and training-time overhead. Reviewers raised concerns about the computational cost of expert queries and the absence of training-time analysis. *(CHBo: Weakness 2; EoSa: Question 2; oegD: Weakness 3; Ka6W: Weakness 3)* We added measurements showing that EAPO requires ~1.4× the training time of self-exploration RL, largely due to expert calls for higher-quality trajectories. However, this overhead is mitigated through on-demand consultation, annealed reduction of expert usage, and no expert calls during inference.
> - All other questions regarding the understanding of the paper have been carefully answered by the reviewers; issues such as formula subscripts have also been corrected in the latest version of the paper.
>
> Above, we have faithfully summarized all reviewer comments and our corresponding responses, hoping that this will assist the AC's work. We are deeply grateful to the reviewers, AC, SAC, and PC, for their dedicated effort and excellent work. The authors offer their sincere respect and appreciation to all involved!
>
> Sincerely,
>
> Authors

---

### Meta-Review · Area_Chair_kcoU · 2026-01-06

**Summary:**

This paper proposes EAPO (Expert-Assisted Policy Optimization), a reinforcement learning framework designed to improve LLM reasoning by allowing the model to actively consult external experts during training, thereby overcoming the challenges of sparse rewards and inefficient exploration. By employing a curriculum that gradually reduces reliance on these consultations, EAPO compels the policy to internalize expert knowledge, enabling the model to function autonomously at inference time with significantly improved performance and stability on mathematical reasoning benchmarks compared to standard RL or distillation baselines.

The authors have responded to most of the concerns that the reviewers raised, such as technical novelty, dependence on expert, computational overhead etc.  In particular, I appreciate the authors for performing additional experiments on tasks per reviewers request. However, I have some additional concerns for the work that remain outstanding, which I will detail below.

**Reviewer Concerns:**

The authors have responded to most of the concerns that the reviewers raised, with fruitful additional experimental results on tasks other than math and with required experiment settings.

However, I have some concerns for the additional results posted in rebuttal:

**(1) The claims in the rebuttal (Expert-assisted workflows on non-math datasets:)**:
> *"We observe that introducing an expert model into the workflow yields **consistent and significant performance gains**..."*

However, looking at the **Average** column for the "Base Model" versus the "Expert-Assisted Workflow":

1.  **Base Model (R1-7B)** achieves an Average score of **41.59**.
2.  **Expert-Assisted Workflow (R1-7B + R1-7B)** achieves an Average score of **39.84**.
    * *Result:* **Performance Decrease** (not a gain).
3.  **Expert-Assisted Workflow (R1-7B + QwQ-32B)** achieves an Average score of **40.37**.
    * *Result:* **Performance Decrease** (not a gain).

Only one configuration (R1-7B + R1-14B) shows a gain (43.78), but even then, the performance on specific tasks like **HumanEval** drops significantly as "stronger" experts are added (from **86.79** baseline down to **66.46** with QwQ-32B).

Therefore, the claim that the method yields "consistent" gains is questionable according to their own reported numbers; in 2 out of 3 cases, the average performance actually got worse.

**(2)  A detailed analysis of the early-stage reward signals**

The columns represent the cumulative proportion of rewards falling below a certain threshold (e.g., score ≤ 0.2, score ≤ 0.3). By definition, the percentage of scores less than or equal to 0.2 must be less than or equal to the percentage of scores less than or equal to 0.3. However, for the EAPO method rows from step 15 onwards, the table shows the opposite:

Row "EAPO 20":
- score ≤ 0.2 is 22%
- score ≤ 0.3 is 21% (Impossible violation)

Row "EAPO 30":
- score ≤ 0.2 is 17%
- score ≤ 0.3 is 16% (Impossible violation)

Row "EAPO 40":
- score ≤ 0.2 is 10%
- score ≤ 0.3 is 9% (Impossible violation)

If "score ≤ 0.3" means "0.2<score ≤ 0.3", then it also viololates
Row "EAPO 5":
- score ≤ 0.2 is 43%
- score ≤ 0.3 is 63%
- 43% + 63% > 100% (Impossible violation)

**(3)  Statistical significance in training**

Another potential limitation of this study that has not been discussed by the reviewers are lack of multiple random seed in training. Based on what’s written in the paper, they do not report using multiple random seeds / repeated training trials for training; the “variance” they discuss/report is tied to evaluation-time sampling and per-problem variability, not multiple training runs. For rollout-based learning / self-exploratory RL, there is a huge chance that training result varies significantly among random seeds, which is well-known in classic RL. To guarantee statistical significance, the authors are encourage to train the student model with multiple random seeds.

**Reviewer Scores:**

It is hard to predict how the reviewers would change their score if they had been able to participate fully in the discussion, because of the questionable results in the rebuttal as mentioned above.

---

### Decision · Program_Chairs · 2026-01-26

Reject